# Measuring Fingerprints of Web-filtered Text Datasets and Fingerprint Propagation Through Training

**Youssef Mansour**
Technical University of Munich
Munich Center for Machine Learning
y.mansour@tum.de

**Reinhard Heckel**
Technical University of Munich
Munich Center for Machine Learning
reinhard.heckel@tum.de

## Abstract

We investigate fingerprints in pretraining datasets for large language models (LLMs) through dataset classification experiments. Building on prior work demonstrating the existence of fingerprints or biases in popular computer vision datasets, we analyze popular open-source pretraining datasets for LLMs derived from Com-monCrawl including C4, RefinedWeb, DolmaCC, RedPajama-V2, FineWeb, and DCLM-Baseline. Despite those datasets being obtained with similar curation steps, neural networks can classify surprisingly well which dataset a single text sequence belongs to, significantly better than a human can. This indicates that small differences in filtering and processing pipelines induce fingerprints, that we find are evident in formatting, vocabulary, and content distributions. Such fingerprints can negatively impact cross-dataset generalization. Additionally, we show that these fingerprints propagate through training: sequences generated by models trained on those datasets can be accurately classified by a classifier trained on the original datasets. This can offer insights into data characteristics that are typically undisclosed by LLM developers, including pretraining mixture proportions and finetuning data sources.

## 1 Introduction

In 2011, Torralba and Efros [TE11] proposed the dataset classification experiment to examine unique fingerprints (or biases) present in common computer vision datasets. The paper demonstrated that computer vision researchers can classify well which dataset an image from a computer vision dataset popular at the time (e.g., PASCAL, Caltech101, ImageNet,...) belongs to. Moreover, classifiers can be trained to relatively reliably classify which dataset an image comes from. While some of the fingerprints can be accounted for by isolating specific objects the different datasets focus on, Torralba and Efros [TE11] found that the fingerprints or biases are still present in some form, even if those effects are isolated.

Recently, Liu and He [LH25] revisited the dataset classification experiment in the current era of large-scale and diverse vision datasets like YFCC [Tho+16], DataComp [Gad+23], and LAION [Sch+22]. Those datasets are collected to train generalizable representations, as opposed to datasets collected for a specific purpose (for example for urban scene understanding [Cor+16]). Liu and He [LH25] found, perhaps surprisingly, that even for those large and diverse datasets, classifiers can relatively accurately assign single images to the datasets.

In this paper we study the fingerprints or biases of popular pretraining datasets for large language models (LLMs), investigate their origin, and show that they propagate through training. We also study how those fingerprints can impact generalization, and can be leveraged to gain information on the training and finetuning data of LLMs.

39th Conference on Neural Information Processing Systems (NeurIPS 2025) Track on Datasets and Benchmarks.

We consider the most popular open web-based datasets for general purpose LLMs, specifically C4 [Raf+20], RefinedWeb [Pen+23], DolmaCC [Sol+24], RedPajama-V2 [Tog23a], FineWeb and FineWeb-edu [Pen+24], and DCLM-Baseline [Li+24]. These datasets consist of sequences of text of average length ranging from 477 to 1235 tokens (see Appendix A), and are obtained by pre-processing and filtering CommonCrawl. These datasets are diverse and are commonly used for pretraining LLMs.

Our main findings are:

- **Distinguishability:** Sequences from popular pretraining text datasets can be well classified to belong to a certain dataset, highlighting unique fingerprints inherent in these datasets. For example, the datasets C4, RefinedWeb, and FineWeb are all obtained from CommonCrawl using similar deduplication and heuristic quality filtering steps. Yet an LLM trained to distinguish between C4, RefinedWeb, and FineWeb achieves 74.8% accuracy, well above chance (33.3% accuracy) and human accuracy.

- **Distinguishing features:** We analyze the features that enable detectability of sequences, and find that the datasets differ in format, vocabulary, and content distributions. However, there is not a single feature on its own that makes the sequences easily distinguishable, and even rewritten sequences with removed formatting are well distinguishable by a classifier.

- **Fingerprint propagation through training:** Sequences generated by models trained on these datasets can be accurately classified to belong to their respective datasets using a classifier trained on the original data.

- **Finetuned models:** Popular LLMs including GPT-4o, Claude-3.5-Sonnet, Qwen-2.5, Gemini-2.0-Flash, LLama-3.3, DeepSeek-V3, and GPT-OSS generate sequences that are generally well distinguishable, but GPT-4o and DeepSeek-V3 generate surprisingly difficult to distinguish sequences depending on the prompt distribution.

Implications of our findings are:

- **Cross-dataset generalization:** The distinguishability of sequences in the datasets suggests that despite the datasets being large and diverse, they contain dataset-specific features. As a result, models pretrained on a single dataset can struggle to generalize to others, as measured by perplexity, as we demonstrate. Jointly training on a mixture of datasets leads to improved generalization measured in perplexity.

- **Mixture proportions estimation:** LLMs pretrained on several data sources can generate random sequences that reflect the proportion of the data sources in the pretraining mixture. By classifying the generated sequences with a classifier trained to distinguish between the original data sources, we can estimate the pretraining mixture proportions.

- **Insights into finetuning data:** Responses from GPT-4o and DeepSeek-V3 to OpenHermes prompts are particularly difficult to distinguish, relative to other LLMs and datasets. This hints at the potential inclusion of GPT-4o-generated responses to OpenHermes prompts in DeepSeek-V3's finetuning data.

In an earlier version of this paper, we used the term bias instead of fingerprints following prior computer vision literature [TE11; ZYL24; LH25]. However, bias can be easily be mistaken for social or stereotypical biases (e.g., gender, race), so we instead adopt the more specific term fingerprints.

## 2 Related work

This work is inspired by Torralba and Efros [TE11]'s dataset classification experiment for vision datasets and Liu and He [LH25]'s recent work that revisited the dataset classification experiment in the context of modern large scale dataset. Liu and He [LH25] found, similar as we find for language datasets, that images from the large scale and diverse computer vision datasets YFCC [Tho+16], CC [Cha+21], and DataComp [Gad+23] can be accurately classified as belonging to one of those datasets. Zeng et al. [ZYL24] extended Liu and He [LH25]'s work, and explored the specific forms of fingerprints present in large-scale vision datasets by performing classification experiments on various transformations of the original datasets. In our work, we identify fingerprints for text datasets, study their origin, propagation, and implications.

A variety of works study the problem of classifying LLM generated text. [Han+24] and many phrase this as a classification problem [Sol+19; Tia+24; HCH23]. Guo et al. [Guo+23] demonstrate that ChatGPT generated answers can be well distinguished from human answers by a classifier, if the text is sufficiently long. In this work we focus on distinguishing popular pretraining text datasets with a classifier, not AI vs human generated text.

Shi et al. [Shi+23] and Maini et al. [Mai+24] consider the problem of detecting pretraining data based on blackbox access of LLMs; specifically given a text and blackbox acccess to an LLM, was the LLM trained on that text? Carlini et al. [Car+21] and Nasr et al. [Nas+23] attempt to extract training data from LLMs. They show that an adversary can extract verbatim text sequences from the model's training data by querying the LLM with no previous information of the training set. In Sec. 5 we study the loosely related problem whether a classifier trained to distinguish training data can distinguish data generated by LLMs trained on it.

Xie et al. [Xie+23] and Ge et al. [Ge+24] optimize the mixture proportions of different data sources for pretraining an LLM to maximize its performance on specific tasks. In Sec. 5.1 we obtain an LLM pretrained on different data sources, and estimate the proportion of each source in the training mixture.

# 3   Setup and datasets considered

Throughout this paper, we perform dataset classification for language datasets as follows. Each dataset consists of a set of sequences, and a classifier is trained to distinguish the sequences from $N$ such datasets. We measure performance on a test set consisting of an equal amount of sequences from each of the $N$ datasets.

As classifier, we use a pretrained autoregressive transformer with 160M parameters, which we finetune on a training set of sequences to perform $N$-way classification. See Appendix B for the details of the model used and training specifications.

## 3.1   Distinguishing data from different sources

LLMs are often pretrained on data from different sources, for example LLama 1's [Hug+23]'s pretraining data, and reproduction of the data, RedPajama-1T [Tog23b], consists of the sources C4, CommonCrawl (CC), Arxiv, Github, Wikipedia, and Stack Exchange. Some of those sources are very easy for humans to distinguish, for example Github (containing code) and C4/CC (containing little code). Thus, it is perhaps not surprising that we find that six-way classification of the Redpajama-1T sources (C4, CC, Arxiv, Github, Wikipedia, Stack Exchange) yields an accuracy of 98.25%.

## 3.2   Datasets considered

We consider seven of the largest and most popular open datasets for pretraining general-purpose LLMs based on web-filtered data: C4 [Raf+20], RefinedWeb [Pen+23], DolmaCC [Sol+24], RedPajama-V2 [Tog23a], FineWeb and FineWeb-edu [Pen+24], and DCLM-Baseline [Li+24]. See Appendix C for a detailed description of each dataset.

The datasets are based on web crawls from CommonCrawl, a nonprofit that provides a publicly available web archive. Much of the text extracted by CommonCrawl is not useful for training, like three-word sentences and HTML artifacts.

All datasets are obtained by i) extracting text using parsers like `resiliparse` or using Common-Crawl's pre-extracted text, ii) applying heuristic filtering (e.g., language filtering, removing very short texts, and removing texts with curly brackets { since those indicate code), iii) deduplication (for example, identical or nearly identical webpages are filtered out), and iv) machine learning based filtering (for example filtering based on a classifier trained to distinguish high quality data from average data). The exact choices of those steps have a significant effect on the composition of the datasets and on the performance of the models trained on them.

All datasets are based on web crawls, are large in scale, and are broad, i.e., not focused on a specific topic or area (such as Arxiv, Github, etc). Therefore it is perhaps surprising that sequences of these datasets can be relatively reliably distinguished.

| # Classes | Category 1 | | | Category 2 | | Category 3 | | Accuracy |
|---|---|---|---|---|---|---|---|---|
| | C4 | FineWeb | RefinedWeb | DolmaCC | RedPajama-V2 | DCLM | FineWeb-Edu | |
| | ✗ | | ✗ | | ✗ | | | 80.50% |
| | ✗ | ✗ | | | ✗ | | | 79.27% |
| | | | ✗ | ✗ | ✗ | | | 77.99% |
| | | ✗ | | ✗ | ✗ | | | 75.74% |
| | ✗ | ✗ | ✗ | | | | | 74.76% |
| | ✗ | | | ✗ | ✗ | | | 74.09% |
| | | ✗ | ✗ | ✗ | | | | 73.04% |
| 3 | ✗ | | ✗ | ✗ | | | | 72.90% |
| | ✗ | ✗ | | ✗ | | | | 68.84% |
| | | ✗ | ✗ | | ✗ | | | 67.55% |
| | ✗ | | | | | ✗ | ✗ | 94.12% |
| | | | | ✗ | | ✗ | ✗ | 92.94% |
| | | | ✗ | | | ✗ | ✗ | 89.76% |
| | | ✗ | | | | ✗ | ✗ | 85.16 % |
| | | | | | ✗ | ✗ | ✗ | 84.55% |
| | ✗ | ✗ | ✗ | | ✗ | | | 70.31% |
| | ✗ | | ✗ | ✗ | ✗ | | | 68.98% |
| 4 | | ✗ | ✗ | ✗ | ✗ | | | 67.88% |
| | ✗ | ✗ | | ✗ | ✗ | | | 67.45% |
| | ✗ | ✗ | ✗ | ✗ | | | | 64.44% |
| 5 | ✗ | ✗ | ✗ | ✗ | ✗ | | | 60.70% |

Table 1: Classification accuracy across different combinations from the three dataset categories. Despite the similarity in the filtering techniques, high classification accuracy is observed, specially for category 3.

# 4 Dataset classification experiments

We group the seven datasets into three categories based on their preprocessing techniques. Category 1 consists of the language filtered, heuristically filtered, and deduplicated datasets C4, FineWeb, and RefinedWeb datasets. Category 2 consists of datasets processed with Category 1 steps and additional light filtering based on Wikipedia perplexity scores, and includes the Dolma and RedPajama-V2 datasets. Category 3 consists of datasets processed with Category 1 steps and carefully selected machine learning-based text filtering techniques and includes DCLM-Baseline and FineWeb-Edu.

We perform all possible combinations of three-way, four-way, and five-way classification using the five datasets from categories 1 and 2. Additionally, we perform five three-way classification experiments that pair the two datasets from category 3 with each of the five datasets from the other categories. We also report the results for all two-way combinations in Appendix D. We train a 160M transformer on 160M training tokens per dataset, i.e., 480M for three-way, 640M for four-way, and 800M for five-way classification. As a test set we take 8192 unseen sequences from every dataset.

As seen in Table 1, across all dataset combinations, the classifiers consistently achieve high accuracy. Particularly high accuracy is obtained when classifying sequences from DCLM-Baseline vs the other datasets, which is perhaps not surprising since those sequences are relatively distinct, see Appendix E for examples.

However, it is perhaps surprising that sequences from the datasets processed with similar language and heurisitc filtering and deduplication steps are easily distinguishable. Humans perform significantly worse in assigning text sequences to datasets, see Section 4.1 below.

In Appendix F we provide ablation studies justifying the choice of our classifier including scaling the training data and model size.

## 4.1 Classification accuracy achieved by humans

Our experiments show that classifiers can accurately differentiate between datasets, even when the differences are subtle to human perception, as seen from the two examples from C4 and FineWeb in Figure 1. More examples are in Appendix E.

We conducted a dataset classification experiment to measure human performance. The task is binary classification between C4 and FineWeb. We gave two machine learning researchers several sequences

| Datasets | Training sets | | | | | | | |
|---|---|---|---|---|---|---|---|---|
| | C4 | FineWeb | RefinedWeb | DolmaCC | RedPajama-V2 | DCLM | FineWeb-Edu | Mixture |
| C4 | **30.5** | 34.2 | 34.2 | 35.4 | 38.0 | 41.2 | 43.5 | 32.8 |
| FineWeb | 39.2 | **34.4** | 35.2 | 40.8 | 39.3 | 41.4 | 45.1 | 34.8 |
| RefinedWeb | 46.1 | 38.9 | **35.1** | 49.8 | 43.1 | 46.2 | 51.9 | 37.4 |
| DolmaCC | 33.0 | 32.9 | 32.9 | **31.9** | 33.8 | 36.9 | 39.6 | 32.0 |
| RedPajama-V2 | 34.8 | 29.5 | 28.9 | 34.9 | **26.7** | 33.1 | 35.7 | 27.2 |
| DCLM | 54.8 | 53.1 | 45.4 | 48.0 | 54.3 | **32.3** | 57.2 | 33.4 |
| FineWeb-Edu | 31.4 | 28.9 | 29.5 | 30.4 | 29.2 | 28.9 | **23.7** | 26.4 |
| Mixture | 37.8 | 35.3 | 34.1 | 38.0 | 36.8 | 36.4 | 40.6 | **31.6** |
| WikiText-103 | 47.3 | 44.6 | 46.0 | 46.5 | 49.0 | 46.3 | 45.4 | **42.7** |
| Paloma | 55.9 | 54.3 | 47.6 | 61.8 | 55.0 | 42.2 | 66.3 | **41.2** |

Table 2: Cross dataset and benchmark generalization in terms of perplexity. For each row, the lowest and second lowest perplexity values are shown in bold and underlined, respectively.

from each dataset for inspection. For testing, the researchers were given 50 unlabeled sequences from each set.

The researchers achieved an average accuracy of 63%, only 13% above random guessing. In contrast, the 160M sized classifier attains 88%, which highlights the model's ability to identify subtle patterns that are not easily distinguishable by humans.

| C4 |
|---|
| •Made it back, can I come inside for a change? Made of glass and falling fast all the way! Thanks for correcting Tokyo Police Club - Miserable lyrics!
•Jamie Oliver is a famous CHEF from the UK. Here you can learn how to make scramble eggs in three different ways: English, French and American way! ENJOY IT! |

| FineWeb |
|---|
| •Short-term and long-term changes in the strength of synapses in neural networks underlie working memory and long-term memory storage in the brain.
•Yesterday, we indulged in all the goodness of sweets, so I thought it only appropriate that we feature the other side of the coin: Salty. Now, I'm a girl who loves her potato chips. |

Figure 1: Sample text sequences from C4 and FineWeb. For a Human, it is difficult to identify patterns to distinguish between the datasets.

## 4.2 Generalization and significance of dataset classification

If two datasets can be reliably distinguished through classification experiments, this suggests that they contain dataset-specific features. Consequently, a model pretrained on one dataset may not generalize well to others, as measured by perplexity. Therefore, when datasets are well distinguishable, like the pretraining datasets considered here, this can suggest that mixing them can improve generalization. To verify this, we pretrain a transformer (with 160M parameters on 3.2B tokens for next token prediction) on each of the seven datasets, as well as on a mixture of them (a total of 3.2B tokens, equally sampled from each dataset).

We measure performance in perplexity, a standard evaluation metric, which measures how well an LLM predicts a sequence of text, and often correlates with performance on real-world downstream tasks [KP02; Gon+23; TPH25]. For evaluation, we sample 1000 unseen sequences from each dataset. We compute the perplexity of each pretrained model on each evaluation set individually, and on the mixture of all evaluation sets. The results are in Table 2.

We also evaluate on two benchmarks: WikiText-103 [Mer+17], which is extracted from the set of verified Good and Featured articles on Wikipedia, and is considered to have high quality texts, and Paloma [Mag+24], a comprehensive benchmark for evaluating the perplexity of LLMs, covering a diverse set of domains, including Twitter, code, Reddit, StackExchange, and academic text, making it a robust measure of generalization across various real-world data sources.

Models pretrained on a single data source have lowest perplexity on that source, but high perplexity on other sources. In contrast, the model trained on the mixture consistently has the second lowest

perplexity on each individual dataset, and the lowest perplexity on the mixture evaluation data, and importantly, also on the benchmarks WikiText-103 and Paloma.

The reduction in perplexity due to mixing data sources, that are well distinguishable by a classifier, highlights the benefit of training on a diverse corpus, which improves the model's ability to generalize across a broader range of language distributions.

### 4.3 Features enabling dataset distinguishability

To gain insights into what makes the sequences distinguishable, we conduct rewrite, reformatting, and dataset categorization experiments. We find that format, vocabulary, and content are all characteristics that enable differentiating between the datasets, but no single feature on its own fully explains the distinguishability. While some of these features are easily identifiable, others are subtle and not easily identifiable by humans.

An easily identifiable example feature is the formatting of DCLM-Baseline. The DCLM team used resiliparse to extract text, which very frequently inserts new lines between the sentences (i.e., ends sentences with \n\n). This makes DCLM sequences particularly distinct, see Appendix E. This is also reflected in the high accuracy a model attains when classifying DCLM sequences as seen in Tables 1 and 5.

Another example is FineWeb-Edu. Most of the sequences are educational and scientific, and thus classifying educational and scientific sequences as FineWeb-Edu sequences can work relatively well, see Appendix E for examples.

#### 4.3.1 Rewrite experiments

We rewrite original data with an LLM and classify the rewritten texts. We rephrase the datasets with GPT-4o-mini prompted with the following three prompts:

**Prompt 1:** "*Rewrite the following text sentence by sentence while preserving its length and the accuracy of its content. Maintain the overall format, structure, and flow of the text:*"

**Prompt 2:** "*Rewrite the following text while preserving its length and the accuracy of its content:*"

**Prompt 3:** "*Rewrite the following text while preserving its length and the accuracy of its content. Do not use newlines, new paragraphs, itemization, enumeration, and other formatting, unless it is important or appropriate for better readability:*"

The prompts encourage increasing degrees of deviation of the rephrased texts from the originals as seen in Appendix G.

We consider the binary classification task between C4 and FineWeb, i.e., we train a 160M transformer to distinguish rephrased C4 from rephrased FineWeb. Using each of the prompts, we rephrase 160M training tokens and 8192 test sequences from every dataset.

The classification accuracy between the original text is 87.4% followed by (i) 83.2% for the text rephrased with Prompt 1, (ii) 79.5% for Prompt 2, and (iii) 66.0% for Prompt 3.

Interestingly, while the rephrased text is more difficult to distinguish, when rewritten with Prompt 1 and Prompt 2, the sequences are still distinguishable for a classifier. This suggests that the distinguishability of the texts does not overly rely on wording. Subtle format differences play a more significant role, as suggested in the large accuracy drop with Prompt 3.

#### 4.3.2 Removing formatting and classifying based on word frequencies only

To gain further insights into the effect of formatting and vocabulary on the datasets' distinguishability, we unify formatting, and classify based on unique word frequencies only.

**Removing formatting** We remove structural formatting of C4 and FineWeb by removing all newlines, itemization and enumeration patterns, including numbers, bullet points and similar markers that commonly denote list elements, excessive spaces, and other special characters such as tabs and carriage returns. The resulting text is a single continuous block of text.

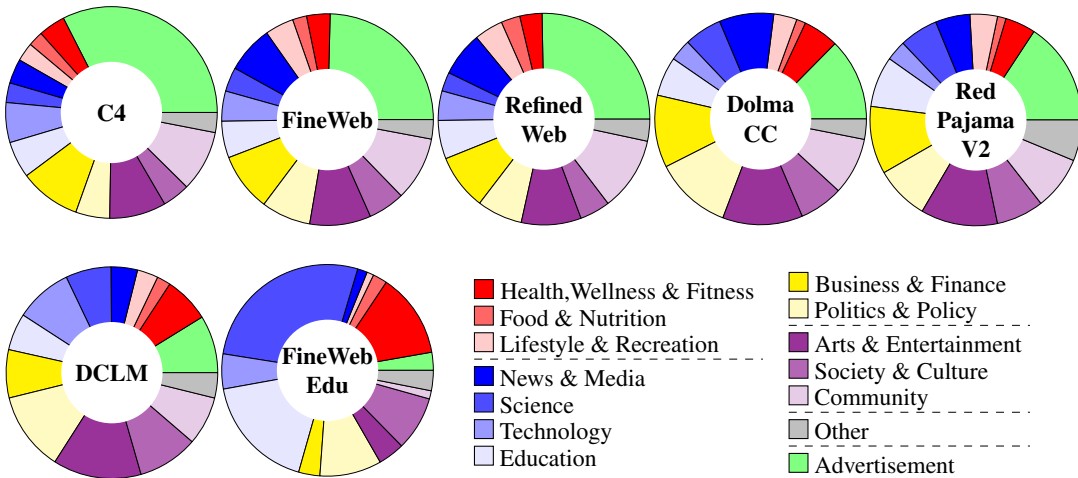

Figure 2: Categorization of datasets into 13 thematic categories. Similarly filtered datasets have comparable categorical distributions.

We train the 160M model to classify between regex preprocessed C4 and FineWeb. We use 320M training tokens (160M tokens per dataset), and 8192 test sequences. The accuracy is 72.42%, about 15% less than the accuracy on the original datasets (87.37%). This drop in accuracy suggests that models detect patterns in formatting that are important for classification. However, the fact that classification remains relatively accurate even after unifying the formatting, suggests that there are fingerprints beyond format and structure.

**Bag of Words** is a simple text classification method that represents text as a collection of unique words, disregarding format, grammar, word order, or context. Each text sequence is transformed into a vector with the frequency of each unique word within the text. For instance, Bag of Words transforms the following two texts: "*I like apples but not bananas*" and "*I like bananas but not apples*" to the same exact vector representation.

We use Bag of Words to distinguish between C4 and FineWeb, and achieve a classification accuracy of 63.45%. Classification with Bag of Words is higher than a random guess despite reducing each text sequence to a vector with the frequency of words within it. Bag of Words disregards any semantic relationship between the words, it is based solely on the vocabulary used, which suggests that the vocabulary distributions of C4 and FineWeb are different.

### 4.3.3  Dataset categorization

To get a deeper understanding of the characteristics that differentiate the datasets, we obtain a random sample from each of the seven datasets, and categorize its text sequences into the 13 thematic categories in Figure 2. We use GPT-4o-mini's API by prompting it to classify the text to the most appropriate category. If none of the categories are appropriate, it chooses "Other".

The results in Figure 2 reveal that the content distribution is close for similarly filtered datasets. For instance C4, FineWeb, and RefinedWeb are filtered with standard heuristics and deduplication, and therefore have a comparable distribution. DolmaCC and RedPajama-V2 are additionally filtered with respect to Wikipedia perplexity and thus also exhibit similar distributions.

The machine learning filtered datasets (Dolma, RedPajama, DCLM, and FineWeb-Edu) have significantly less advertisement content than C4, FineWeb, and RefinedWeb. Also, FineWeb-Edu is filtered for educational content, and therefore has many sequences categorized as "Science" and "Education". Such content differences across datasets provide a basis for distinguishability.

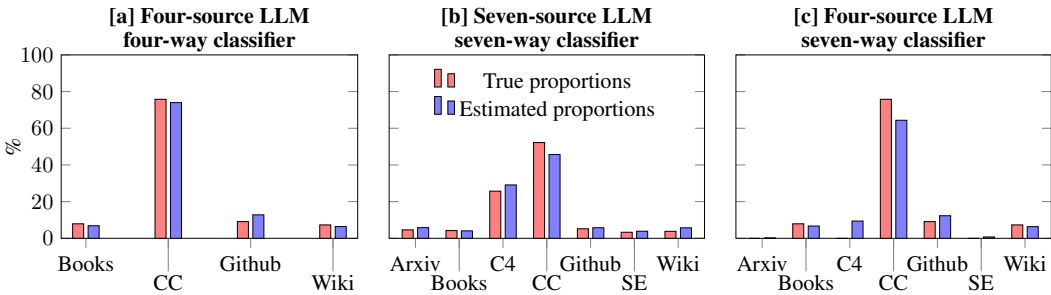

Figure 3: Percentage of generated sequences assigned to different data sources by a classifier trained on original data. **[a]** Sequences generated by an LLM trained on four sources and classified by a classifier trained on the same four sources. **[b]** Same as [a] but seven sources. **[c]** Sequences generated by an LLM trained on four sources and classified by a classifier trained on seven sources.

## 5    Fingerprint propagation

We now explore how the fingerprints inherent in the datasets propagate to text generated by LLMs trained on those datasets. To this end, we evaluate how well a classifier trained to distinguish the original data can classify the generated data.

We consider the following three publicly available LLMs pre-trained on individual datasets from the seven datasets considered in this study:

- Falcon-7B: A 7B parameter model pretrained on 1.5 trillion tokens from the RefinedWeb dataset by TII (Technology Innovation Institute) [Alm+23].
- DCLM-7B: A 7B model pretrained on 2.5 trillion tokens from the DCLM-Baseline dataset by the DCLM team [Li+24].
- FineWeb-Edu-1.8B: A 1.8B parameter model trained by Huggingface on 350 billion tokens from the FineWeb-Edu dataset.

All LLMs are pretrained base models that are not instruction finetuned. See Appendix H for the propagation of fingerprints in instruction-finetuned models.

We generate 8192 test sequences from each LLM by prompting the LLM with a single token, sampled from the distribution of tokens that appear as the first token in the sequences derived from the original training data of the LLM. See Appendix I for sample generated sequences.

Using the three-way classifier trained on the original RefinedWeb, DCLM-Baseline, and FineWeb-Edu data (as described in Sec. 4), we classify the generated data. The classifier achieves 89.15% accuracy on the generated data, only 0.61% less than the accuracy on the original data (89.76% as in Table 1). This indicates that the unique fingerprints inherent in pretraining datasets propagate through training, and can be measured surprisingly well from the outputs of models trained on those datasets.

### 5.1    Estimating mixture proportions

We next show how the fingerprint propagation can be utilized to roughly estimate the mixture proportions of pretraining datasets of an LLM.

LLMs are typically pretrained on a mixture of datasets with certain mixture proportions. These proportions impact model performance and are non-trivial to optimize [Xie+23; Alb+23; Ge+24]. LLM developers often do not disclose the training data and mixture proportions.

We hypothesize that an LLM pretrained on multiple datasets, when prompted with a random token, will generate sequences that closely follow the proportions of its training mixture, since LLMs learn the underlying data distribution during training [Del+24], and generate tokens by sampling from the probability distribution of the learned patterns.

To verify this hypothesis, we utilize SlimPajama [She+24], a refined version of RedPajama-1T [Tog23b]. SlimPajama consists of seven data sources: Arxiv, Books, Github, C4, CC (Common

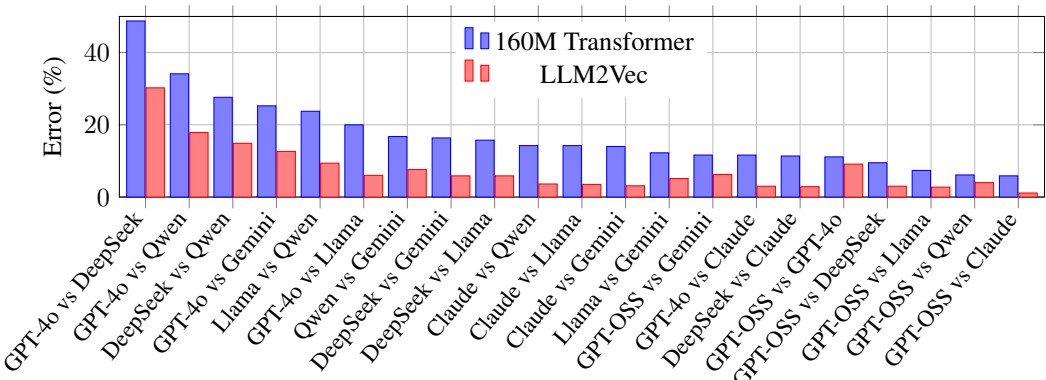

Figure 4: Two-way classification error for distinguishing text generated from popular LLMs by prompting them with OpenHermes-2.5 prompts.

Crawl), SE (Stack Exchange), and Wikipedia. The SlimPajama team provides two 1.3B LLMs trained on 330B tokens from the SlimPajama dataset. The first LLM is trained on only four sources: Books, Github, CC, and Wikipedia, and the second one is trained on all seven sources. The mixture proportion of each source is known.

We train a four-way classifier on the original data from the four sources: Books, Github, CC, and Wikipedia, and another seven-way classifier on the original data from all seven sources. We use the 160M model as a classifier, and 160M training tokens from each source. We generate 2048 random sequences from each LLM by prompting the LLM with one random token, then classify the generated sequences using the the classifiers trained on the original data.

We classify the sequences generated by the LLM trained on four and seven sources using the four-way and seven-way classifiers respectively, and report the percentage of sequences classified as belonging to one of the sources in Figure 3 **[a,b]**. The estimated proportions approximate the true proportions well across most sources. However, the estimates of C4 and CC slightly deviate from the true ones as seen in **[b]**, as some CC sequences are classified as C4. This is somewhat expected as C4 is a subset of CC.

In Figure 3 **[c]**, we use the seven-way classifier to classify the sequences generated by the LLM trained on the four sources, to verify if the classifier correctly refrains from assigning sequences to the 3 excluded sources: Arxiv, C4, and SE. Almost no sequences were classified as Arxiv or SE, confirming that the LLM has not been trained on any of them. As observed previously, a discrepancy appears with some CC sequences misclassified as C4.

## 6  Distinguishability of sequences from popular instruction finetuned LLMs

We now investigate the distinguishability of text generated by popular instruction finetuned LLMs including GPT-4o, Gemini-2.0-Flash, Claude-3.5-Sonnet, DeepSeek-V3 [DA+24], Qwen-2.5-72B [Qwe+25], Llama-3.3-70B [Gra+24], and GPT-OSS-20B [Ope+25].

Generating random sequences with those models is challenging, as directly prompting them with a single token (as we did for pretrained models) results in responses consistent with their task as assistants, for example "*Hello, how can I help you?*".

Instead, we prompt each of the models with the prompts from OpenHermes-2.5 [Tek23], a popular and broad instruction finetuning dataset. We generate 10k responses (about 5M tokens) for training and 400 test responses with each LLM. The performance of a 160M-paramter transformer model trained for pairwise classification of all pairs is in Figure 4.

It can be seen that for the 160M classifier, all LLM pairs are relatively well distinguishable, apart from DeepSeek-V3 and GPT-4o, that are essentially indistinguishable for the classifier (48.7% error rate). This is consistent with the anecdotal evidence that DeepSeek-V3 provides very similar responses to GPT-4o (Appendix J), and hints at part of the instruction finetuning data being generated by GPT-4o.

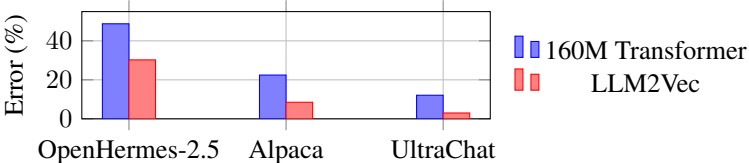

Figure 5: Classification error between texts generated by GPT-4o and DeepSeek-V3 using prompts from three datasets: OpenHermes-2.5, Alpaca, and UltraChat using two classifiers.

To see whether the sequences remain essentially indistinguishable for a much stronger classifier, we train LLM2Vec [Beh+24], which finetunes a Llama-8B-Instruct model using bidirectional attention, and performs very well at higher computational cost. The classification error with this classifier drops to 30%, demonstrating that for an excellent classifier, the responses from DeepSeek-V3 and GPT-4o are distinguishable, but interestingly much less so than sequences from the other models.

To investigate the distinguishability of GPT-4o and DeepSeek-V3 generated sequences further, we now prompt GPT-4o and DeepSeek-V3 with prompts from Alpaca [Tao+23] and UltraChat [Din+23], two popular instruction finetuning datasets. In a recent paper [Sun+25] that appeared after the first version of this paper, the authors carried out dataset classification experiments as well for responses to UltraChat prompts of different LLMs.

The classification accuracy for the 160M transformer and for LLM2Vec shows that GPT-4o and DeepSeek-V3 responses to Alpaca and UltraChat prompts are significantly more distinguishable relative to OpenHermes, see Figure 5.

A hypothesis why DeepSeek-V3 and GPT-4o responses are difficult to distinguish is that OpenHermes prompts with GPT-4o generated responses were part of the finetuning data used for DeepSeek-V3.

Finetuning a model on GPT-4o distilled responses can indeed make the sequences generated by such a model less distinguishable from GPT-4o: We construct a supervised finetuning dataset by prompting GPT-4o with 30k prompts from OpenHermes-2.5 and collecting its responses. We then finetune Qwen-2.5-7B-Instruct on this GPT-4o-generated dataset.

Using OpenHerms prompts distinct from the 30k finetuning prompts, we generate training and evaluation data from both the original Qwen-2.5-7B-Instruct as well as its variant finetuned on the GPT-4o-generated dataset. We train LLM2Vec to distinguish between (i) the original Qwen and GPT-4o, and (ii) the finetuned Qwen and GPT-4o.

The classification accuracy between the original Qwen and GPT-4o is 97.4%, whereas for the finetuned Qwen, it drops by about 20% to 79.8%. This demonstrates that, unsurprisingly, distillation makes sequences from the student and teacher model less distinguishable. These findings highlight the potential of dataset classification experiments in providing insights into the finetuning data of popular LLMs.

# 7   Conclusion

In this work, we demonstrated that popular web-filtered pretraining datasets possess unique and measurable fingerprints, despite their similar origins and curation methods. Through classification experiments, we showed that neural networks can identify a sequence's source dataset with surprisingly high accuracy, a task at which humans perform poorly. We identified that these fingerprints stem from subtle distinctions in formatting, vocabulary, and content distributions. Moreover, we found that these fingerprints propagate through pretraining and finetuning.

# Acknowledgements

The authors are supported by the German Federal Ministry of Education and Research, and the Bavarian State Ministry for Science and the Arts. The authors of this work take full responsibility for its content.

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

## Reproducibility

This work is fully reproducible, as all resources and tools used are publicly available. Our classifier is based on the langue model code from the OpenLM repository [Gur+23], which is a public repository designed for research on medium-sized language models. All datasets considered in this study are publicly available. The rewriting (Sec. 4.3.1) and dataset categorization (Sec. 4.3.3) experiments are performed with GPT-4o mini, and thus leverage the OpenAI API. For our fingerprint propagation experiments we use publicly available pretrained LLMs. For the instruction finetuned LLMs (Sec. 6), we use the respective APIs for the closed-source models (GPT-4o, Claude, Gemini) and the Together AI API for the open-source ones (Llama, Qwen, DeepSeek, GPT-OSS). All code, dataset and LLM download links, and reproduction instructions are on Github: https://github.com/MLI-lab/LLM_data_bias .

We ran all the experiments on A100 GPUs. Our default configuration for the 3-way classification experiment with 480M training tokens and the 160M transformer takes about 75 minutes on one A100 GPU. Pretraining the 160M transformer compute optimal on 3.2B tokens, takes 2.5 hours on four A100 GPUs.

## Limitations

In this paper we demonstrated that popular pretraining text datasets for LLMs contain inherent fingerprints that propagate through training, enabling a classifier trained on original data to effectively classify generated data and, consequently estimate the pretraining mixture proportions. We showed that classification is possible under various conditions such as rephrasing and finetuning.

However, one case where classification accuracy is severely degraded is when datasets consist of the same data sources but differ solely in their mixture proportions. Consider two perfectly distinguishable dataset sources, $\mathbf{A}$ and $\mathbf{B}$. Two datasets $\mathbf{X}$ and $\mathbf{Y}$ are constructed with different mixtures of $\mathbf{A}$ and $\mathbf{B}$, where $\mathbf{X}$ has a higher proportion of $\mathbf{A}$ than $\mathbf{Y}$, and $\mathbf{Y}$ has a higher proportion of $\mathbf{B}$ than $\mathbf{X}$. Sequences from $\mathbf{A}$ in $\mathbf{Y}$ may be misclassified as belonging to $\mathbf{X}$, since $\mathbf{X}$ has seen more sequences from $\mathbf{A}$. Similarly, sequences from $\mathbf{B}$ in $\mathbf{X}$ are likely to be misclassified as originating from $\mathbf{Y}$. This setup highlights how classification becomes unreliable when datasets differ only in data source proportions rather than content or filtering techniques.

## A    Dataset statistics

The datasets we consider in this paper consist of millions to billions of sequences with varying lengths. In this section, we present a statistical analysis on the sequence lengths of the seven datasets. To obtain representative statistics, we randomly sample 100,000 sequences from each dataset and tokenize them with the GPT-NeoX tokenizer. The statistics of the lengths of the tokenized sequences are summarized in Table 3 and histograms are in Figure 6.

| Dataset | Mean | St. Deviation | Mode | Median | Range |
|---|---|---|---|---|---|
| C4 | 477 | 823 | 58 | 253 | 31188 |
| FineWeb | 700 | 1540 | 129 | 410 | 118422 |
| RefinedWeb | 624 | 1549 | 82 | 314 | 137104 |
| DolmaCC | 825 | 1647 | 96 | 451 | 132310 |
| RedPajama-V2 | 1137 | 3191 | 12 | 603 | 274814 |
| DCLM-Baseline | 1235 | 2600 | 101 | 665 | 153768 |
| FineWeb-Edu | 1059 | 1993 | 261 | 597 | 120240 |

Table 3: Statistics of the sequence lengths (in number of tokens) of the seven main datasets considered in this paper.

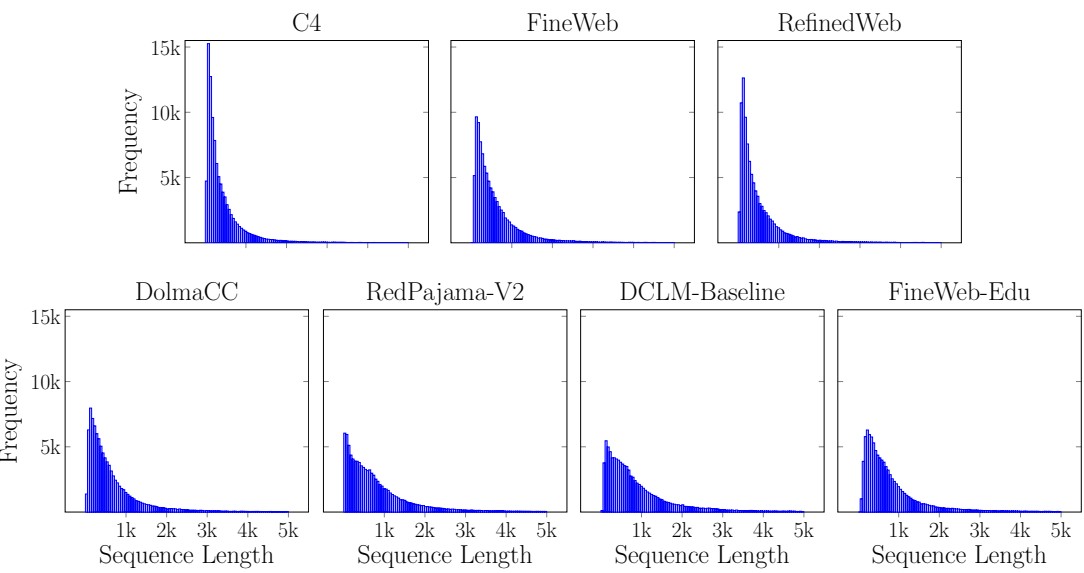

Figure 6: Histograms of the sequence lengths of the main datasets considered. Lengths exceeding 5000 tokens are omitted for ease of visualization.

# B    Model, training details, and hyperparameters

In this section, we detail the architecture of the classifier we use throughout as well as the training procedure and hyperparameters. For all experiments, we utilize the GPT-NeoX tokenizer [Bla+22], which has a vocabulary size of 50,432 tokens.

## B.1    Model

Our primary classifier is a 160M transformer model that we pretrain (for next token prediction) compute optimally [Hof+22] on 3.2B tokens from C4. The C4 data used for pretraining is disjoint from the C4 data used for classification in all other experiments. The same pretrained classifier is used for all classification experiments. After pretraining, we adapt the transformer for our classification tasks, by replacing the final layer with a classification head, similar to the reward model in RLHF [Ouy+22]. Specifically, the original last layer, a linear transformation that maps from the embedding dimension to the vocabulary size, is substituted with a classification head. This classification head is a linear layer that maps from the embedding dimension to $N$, where $N$ represents the number of classification classes.

Additionally, we conduct ablation studies using models of sizes 25M, 87M, and 410M parameters. All models are standard autoregressive transformers, with parameters provided in Table 4.

| Model | 25M | 87M | 160M | 410M |
|---|---|---|---|---|
| Embedding dimension | 192 | 488 | 768 | 1024 |
| Num. heads | 12 | 12 | 12 | 16 |
| Num. layers | 12 | 12 | 12 | 24 |
| Context length | 2048 | 2048 | 2048 | 2048 |
| Vocab. size | 50432 | 50432 | 50432 | 50432 |
| MLP ratio | 8/3 | 8/3 | 8/3 | 8/3 |
| Activation | SwiGLU | SwiGLU | SwiGLU | SwiGLU |
| Weight tying | no | no | no | no |

Table 4: Model parameters. All models have the same architecture, and differ only in the embedding dimension, number of heads, and number of layers.

### B.2 Training and testing details

To prepare the training data, we follow the standard procedure for LLM pretraining. We first tokenize the text sequences using the GPT-NeoX tokenizer. We then construct input sequences of length 2048 tokens, corresponding to the model's context length, by appending sequences of the same dataset together.

An $< |\text{endoftext}| >$ token is added at the end of every sequence before concatenating it with the subsequent sequence. The resulting training sequences, each of length 2048, are partitioned into shards. Each shard contains 8192 sequences, resulting in a total of $8192 \times 2049 = 16.78\text{M}$ tokens per shard.

We train the transformer with a classification head to classify which dataset a text sequence is coming from using the cross-entropy loss. The loss is computed at the token level, where the model classifies every sub-sequence within a given sequence. For instance, a sequence of length 2048 tokens is seen by the model as a series of sub-sequences of lengths 1, 2, 3, ..., 2047, and 2048. Each sub-sequence is classified individually under the same class as the original sequence, ensuring that the model learns to predict the class consistently across all sub-sequence lengths.

At test time, the text sequences are tokenized and fed into the model in their original form, without concatenation. Consequently, the test sequences vary in length. If a sequence originally exceeds 2048 tokens, the model processes only the first 2048 tokens, as this is its maximum context length. Unlike the training phase, where sub-sequences are classified, the model classifies the entire sequence as a whole during testing.

### B.3 Hyperparameters

In all experiments, we train each model for a single epoch, which means that each training token is seen by the model only once. We use a batch size of 16 and apply gradient clipping with a norm of 1 to stabilize training. The initial learning rate is set to 0.0003 and is decayed to zero using a cosine annealing scheduler, with a warm-up phase of 2000 steps.

The optimizer used is AdamW [KB15] with hyperpameters: $\beta_1 = 0.9$, $\beta_2 = 0.95$, $\epsilon = 1 \times 10^{-8}$, and weight decay 0.2. We also use automatic mixed precision training with brain floating point 16 (bfloat16) to enhance computational efficiency throughout the training process.

## C Description of the datasets

In this section we provide a description of the seven main datasets considered in the paper:

**C4**: The Colossal Clean Crawled Corpus [Raf+20] is a popular dataset consisting of 360B Tokens obtained from CommonCrawl text extracted in April 2019, followed by i) language filtering, ii) heurisitc filtering, and iii) deduplication.

**FineWeb:** FineWeb [Pen+24] is a 15T token dataset extracted from CommonCrawl through i) language and ii) heuristic quality filtering and iii) deduplication. The heuristic filters and deduplication steps are carefully chosen based on ablation studies.

**RefinedWeb:** RefinedWeb [Pen+23] is a large scale (5T tokens, 600B publicly available) obtained from CommonCrawl by i) language and ii) heuristic filtering and iii) deduplication.

**Dolma CC:** Dolma [Sol+24] is an open corpus of 3T tokens from different sources. The biggest proportion, about 2.4T tokens, is obtained from CommonCrawl. We consider the CommonCrawl part, which was obtained by first downloading about a quarter of the most recent CommonCrawl data in 2023 (i.e., data from 2020-05 to 2023-06), and was processed with i) language and ii) heuristic quality filtering and iii) deduplication. As a machine learning based quality filtering step, for each sequence the perplexity was computed to measure Wikipedia-likeness (following the CCNet pipeline [Wen+20]) and partitioned into head, middle, and tail by perplexity; we consider the head and middle parts.

**RedPajama-V2:** RedPajama-V2 [Tog23a] is a corpus of 30T filtered and deduplicated tokens also processed with i) language and ii) heuristic quality filtering, and iii) deduplication. We consider the 20.5T token part of the corpus consisting of English speaking documents, as for all other datasets we also consider the English part only. The data contains a broad coverage of CommonCrawl, and

comes with quality annotations that enables slicing and filtering the data. As a machine learning based quality filtering step, for each sequence the perplexity was computed to identify Wikipedia-like documents and partitioned into head, middle, and tail, and head and middle was retained. We consider head and middle as for Dolma CC.

**DCLM-Baseline:** DCLM-Baseline [Li+24] was obtained from CommonCrawl through i) text extraction with resiliparse and language and ii) heuristic quality filtering, deduplication, and iv) machine learning based quality filtering. All steps where chosen by ablation studies to obtain a dataset so that models trained on it perform well. The final, machine learning based filtering step is important and is trained to classify instruction-formatted data from OpenHermes 2.5 and high-scoring posts from the r/ExplainLikeImFive subreditt from RefinedWeb. Models trained on this dataset perform very well on common benchmarks.

**FineWeb-Edu:** FineWeb-Edu [Pen+24] is obtained from FineWeb through machine learning based quality filtering to obtain data with educational text, and consists of 1.3T tokens. Models trained on FineWeb-Edu perform very well on knowledge and reasoning benchmarks such as MMLU [Hen+21].

# D   Two-way classification

Our main results in Table 1 are for three-, four-, and five-way classification between the main datasets considered in this study. In this section, we report the classification accuracy for all possible binary combinations between the seven datasets, i.e., $\binom{7}{2} = 21$ possible combinations.

As before, we use the 160M model with 160M training tokens and 8192 test sequences per dataset. The results are in Table 5.

|  | C4 | FineWeb | RefinedWeb | DolmaCC | RedPajama-V2 | DCLM | FineWeb-Edu |
|---|---|---|---|---|---|---|---|
| C4 |  | 87.37% | 90.72% | 69.42% | 95.64% | 98.85% | 92.88% |
| FineWeb | 87.37% |  | 75.49% | 82.70% | 80.54% | 99.15% | 78.05% |
| RefinedWeb | 90.72% | 75.49% |  | 88.32% | 80.68% | 99.03% | 84.74% |
| DolmaCC | 69.42% | 82.70% | 88.32% |  | 90.91% | 97.03% | 91.08% |
| RedPajama-V2 | 95.64% | 80.54% | 80.68% | 90.91% |  | 99.05% | 77.69% |
| DCLM | 98.85% | 99.15% | 99.03% | 97.03% | 99.05% |  | 98.54% |
| FineWeb-Edu | 92.88% | 78.05% | 84.74% | 91.08% | 77.69% | 98.54% |  |

Table 5: Classification accuracy for all possible two-way combinations of the seven main datasets in this study.

# E   Original sequences

We display sequences from the original seven main datasets considered in this study: C4, FineWeb, RefinedWeb, DolmaCC, RedPajama-V2, DCLM-Baseline, and FineWeb-Edu. A detailed description of the creation of those datasets is in Appendix C.

For clarity and ease of visualization, only short sequences are shown here. Some sequences are considerably longer and span multiple pages. Therefore, the sequences shown here do not reflect the average sequence length.

## C4

• Beginners BBQ Class Taking Place in Missoula!
Do you want to get better at making delicious BBQ? You will have the opportunity, put this on your calendar now. Thursday, September 22nd join World Class BBQ Champion, Tony Balay from Lonestar Smoke Rangers. He will be teaching a beginner level class for everyone who wants to get better with their culinary skills.
He will teach you everything you need to know to compete in a KCBS BBQ competition, including techniques, recipes, timelines, meat selection and trimming, plus smoker and fire information.
The cost to be in the class is $35 per person, and for spectators it is free. Included in the cost will be either a t-shirt or apron and you will be tasting samples of each meat that is prepared.

• Hurrah! A cooperative worldwide effort to rescue Thailand children trapped in a flooded cave rescued them all in less than 3 weeks from the time they entered the cave to the time of their rescue.
It should be much easier, shouldn't even take a heroic effort, to rescue children trapped in separation from their families at the Mexican border. These things are possible, but this week, the administration did not even meet the first deadline to get all the children below 5 years old reunited with their families.
It should even be logistically possible with a cooperative world wide effort to develop economic systems that could rescue all the hungry children everywhere living in poverty.
In the U.S. alone, 1 in 5 children live in poverty, according to a recently released United Nations report.

## FineWeb

• Originally Posted by bradhs
The only thing you can do is create a Search and Save it with a shortcut key. I do this when I only want to see my corporate email.
1. Go into your Messages and select Search.
2. Set the Service option to the Enterprise Email.
3. Save the Search. Give it a name and a shortcut key.
Use the shortcut key to restrict the email list to only Enterprise email.
Hm. This isn't working for me... When I initiate the search, it comes back with no messages, and I do have some that it should show... The service option choices are: All Services, my pop email address, and Desktop. I selected Desktop. That right?

• I have just updated my TV and Blu ray player but not my amp.
I didn't want to update my Sony STR-DG820 amp because it works so well, but I did want to keep my options open for playing 3D discs so I got the Panasonic DMP-BDT310 because it has two HDMI ports and could route sound through the amp and picture to the TV.
I've gone through every setup and I'm not getting DTS-HD or TRUE-HD. The manual doesn't help at all and this is becoming a little silly. Could some one go through a step by step guide in the setup.

## RefinedWeb

• A huge thank you goes to those who helped with the hedge cutting on the road side of the churchyard recently. This was a very much needed task, the pathway is nice and clear now. We are also very grateful to whoever donated the funds to provide the skip, again this was a much needed requirement.
If anyone is interested in helping to maintain the churchyard, please contact Mr Mike McCrea on 01283 214473. Any assistance will be gratefully received.

• Free US shipping on orders over $50!
Pumpkin dominates the fall fragrance scene! This best seller combines brown sugar, molasses, vanilla, and classic holiday baking spices to make an aroma that is simply irresistible!
——!
Amy's review:
"I love these candles. So clever that they're in a coconut shell! The scents fill my house and they have a long burn time! I've purchased from them twice and will continue to support this business! Can't wait to go home and try my fall scents!"

## DolmaCC

• Wowed by the lights and prospects of city life, Loveness leaves her small mining town in search of a new life in Harare. She imagines herself falling for a hot-shot city man becoming his wife and spending her life in luxury while tending to her city children. The man she considers the love of her life is anything but a hot shot, and he is abusive and uncaring. To top all this off, he his HIV positive. Loveness is at a crossroads. She must consider her choices. Although, Waste Not Your Tears does not shy away from misfortune, it is also a novel of forgiveness and hope. Loveness is an unlikely heroine on a stage set during the crisis of HIV/AIDS in Zimbabwe. She lives, however, amongst us, and reading this sensitive and thoughtful novel provides insights into the challenges of making the wrong choices, but having the strength to move forward.

• The Avon Lake Sports Hall of Fame's purpose is to give lasting recognition to the outstanding sports figures and/or teams of Avon Lake who have demonstrated outstanding athletic ability at the high school, college, amateur or professional sports levels.
We strive to recognize those individuals who have contributed greatly to the promotion of sports through leadership, sponsoring, coaching or providing assistance to athletes of athletic programs.
It is our utmost desire to promote more interest in the athletic programs of Avon Lake.

## RedPajama-V2

• Updaty posty thingy Sooo....

Chainmaille was a disaster. I need someone to show me how to construct. It was moot anyway, as I had an anxiety attack and barely made it in to the con. I am so embarased, but glad it wasn't a long term thing.

I am still working on getting the chaim maille done. Maybe it will look fine. I don't know. I also need to work on the scale spoon maille.

Right now, though, my main focus is finding a job. I thought I had extended unemployment until I was done wtih school. Turns out that was not entirely true, and now I am kind of up a creek. I have been saving money, so right now I have been paying bills with my savings. However that is also about to run out. I have applied for abawd.

• Brooklyn Man Who Stabbed 75-Year-Old Woman and Left Her for Dead Sentenced to 75 Years in Prison

Brooklyn Man Who Stabbed 75-Year-Old Woman and

Left Her for Dead Sentenced to 75 Years in Prison

Defendant, a Friend of the Victim's Grandson, Forced His Way into Apartment

Brooklyn District Attorney Ken Thompson today announced that a Brownsville man has been sentenced to 75 years in prison following his conviction on second-degree attempted murder and other charges for stabbing an elderly woman repeatedly and leaving her seriously injured on her apartment floor.

District Attorney Thompson said, "This defendant savagely stabbed a defenseless 75-year-old woman all over her body, robbed her of what little money she had and then left her to die. He deserves every day of his 75-year prison sentence."

## DCLM-Baseline

• Economic Indicators for Libertarians 101

Why Ron Paul is Unique? (Galvanizers and Diplomats)

Ron Paul is a unique figure in libertarianism, able to not only be a diplomat and figure that people outside of libertarianism can empathize with, but also a diehard who can galvanize the most radical of libertarians. It's very rare a figure like him can exist, and let's be glad he does.

• Tuesday, 26 April 2011

tea parties, wonderland, high tea, garden party

I want to hold a cute girly tea party and everyone has to wear their sunday best. I just love the idea. of pretty pastel colours, cupcakes, cooking for your girls and everyone looking pretty

1. This is a beautiful Idea, Ive always wanted to host a tea party and these photos have inspired me to actually go through with it.

2. Beautiful! Could you tell me where you got your cart from? I'm trying to create something similar and they're deceivingly hard to find...

Thankyou for commenting! x

• A "magic" herb, Carissa Edulis, that drew thousands of people to a remote Loliondo village in Tanzania was identified by Kenyan scientists a few years ago as a cure for a drug-resistant strain of a sexually transmitted disease, gonorrhoea. This herb also is believed to cure many other diseases besides gonorrhoea. The Kamba refer to as mukawa or mutote and use it for chest pains, while the Nandi boil the leaves and bark to treat breast cancer, headache and chest pains.

Researchers discovered the plant could be used for the treatment of the herpes virus. Led by Dr Festus M Tolo of the Kenya Medical Research Institute (Kemri), the team from the University of Nairobi and the National Museums of Kenya found the herb could provide an alternative remedy for herpes infections.

"An extract preparation from the roots of Carissa edulis, a medicinal plant locally growing in Kenya, has exhibited remarkable anti-herpes virus activity for both wild type and drug resistant strains," they reported in the Journal of Ethnopharmacology.

• Dinosaurs' active lifestyles suggest they were warm-blooded
H. Pontzer, V. Allen, J.R. Hutchinson/PLoS ONE
Whether dinosaurs were warm-blooded or cold-blooded has been a long-standing question in paleobiology. Now, new research on how two-legged dinosaurs walked and ran adds new evidence to the argument for warm-bloodedness, and suggests that even the earliest dinosaurs may have been warm-blooded.

Warm-blooded (or endothermic) dinosaurs — able to regulate their own body temperatures — would have been more active and could have inhabited colder climates than cold-blooded (or ectothermic) dinos, which would have functioned more like modern reptiles — animals that become animated only as temperatures warm. Endothermic dinosaurs would have also required more energy to maintain their higher metabolic rates.

## F Ablation studies

In this section we perform ablation studies justifying the choice of our classifier. The ablation studies are performed on the three-way classification of C4, FineWeb, and RefinedWeb. Unless stated otherwise, we use the default 160M model with 480M training tokens (160M per dataset) for every ablation study.

We start by scaling the model size, pretraining data, and training data. We find that high accuracy is obtained with different model sizes and dataset set sizes.

**Scaling model and pretraining data:** The default model has 160M parameters and is pretrained on 3.2B tokens. We study the impact of the model size by considering the model sizes 25M, 87M, 160M, and 410M pretrained compute optimally on 0.5B, 1.7B, 3.2B, and 8.2B tokens, respectively. The finetuning set size is kept constant at 480M tokens.

The results are in Figure 7, left panel. For the model sizes considered, the model size and pretraining data amount play a relatively insignificant role; the difference in classification accuracy between the smallest and largest model is only 0.56%.

**Scaling classification training data:** The default training set size used to finetune the pretrained model is 480M tokens. In this study we consider the 160M model pretrained with 3.2B tokens, and finetune it for classification with training sets of different sizes. We start with a training set size of 60M tokens, and then double it up to 1.92B tokens, i.e., we consider the following sizes: 60M, 120M, 240M, 480M, 960M, and 1.92B.

The results are in Figure 7, right panel. The accuracy initially significantly increases with the training data, but saturates close to 480M, which is our default training set size. Quadrupling the training data from 480M to 1.92B tokens only gives a gain of 0.82% in accuracy.

**Accuracy vs sequence length**: As seen in Appendix A, the sequence length varies a lot between the datasets, and even within a given dataset. To evaluate the impact of sequence length on classification accuracy, we analyze sequences of lengths ranging from 0 to 2000 tokens, and divide them into

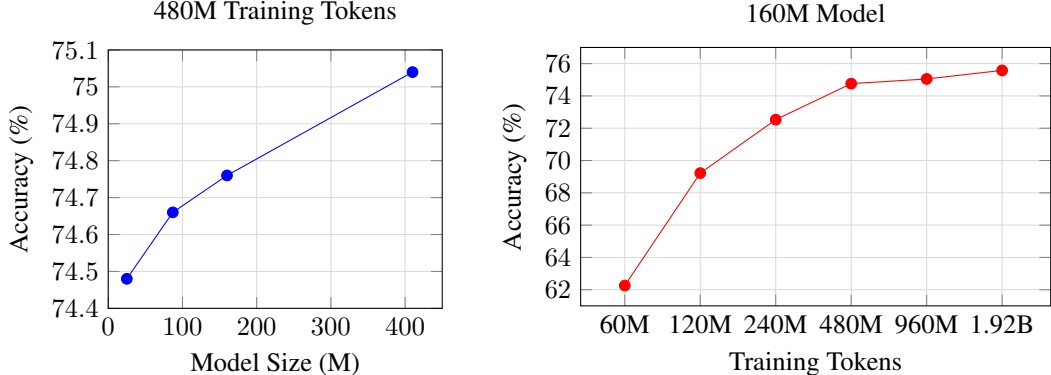

Figure 7: **Left:** Scaling model size and pretraining data with constant training data. **Right:** Scaling training data with constant model size and pretraining data. Scaling model size and pretraining data has a minimal effect on the accuracy, but the effect of the training data is more prominent.

intervals of 200 tokens (i.e., 0-200, 200-400, ..., 1800-2000). For each interval, we sample 1024 test sequences from each dataset.

The results, illustrated in Figure 8, show a steady improvement in classification accuracy as sequence length increases. This trend aligns with the expectation that longer sequences contain more information (2000 tokens is around 1500 words), which allows the classifier to identify more distinguishable patterns and improve classification performance. However, even short sequences can perhaps surprisingly be classified well.

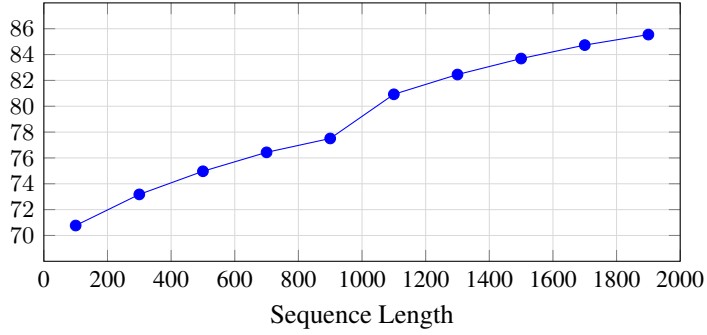

Figure 8: Classification accuracy vs sequence length. Longer sequences attain higher accuracy than shorter ones.

**Training without pretraining:** All classification experiments are carried out by finetuning a model pretrained to predict the next token. To study the impact of pretraining for classification accuracy, we train a randomly initialized model (without any pretraining) directly for classification. This gives an accuracy of 71.59%, which is 3.17% less than the pretrained model (74.76%). Since pretraining improves performance by 3.17%, which is significantly more than increasing the model size, we choose to work with the pretrained model throughout all our experiments.

**LLM2Vec:** LLM2Vec [Beh+24] is a powerful text encoder that also excels as a text classifier. In Sec. 6, we saw that it significantly outperforms the 160M transformer when training data was limited (5M tokens per dataset). In a setup with more data (80M tokens per dataset), the 160M transformer achieves 73% on the three-way classification task of C4, FineWeb, and RefinedWeb. LLM2Vec reaches 82%, still demonstrating superior performance at the cost of increased compute, as it finetunes parameters of a Llama-8B model.

**BERT:** Next we use BERT as a classifier. Unlike autoregressive transformers, BERT [Dev+19] is a bidirectional transformer model that captures contextual information from both preceding and succeeding tokens within a sequence, without the use of causal masks that limit attention to preceding tokens. As a result, BERT processes the entire sequence at once during training, rather than treating it

as a series of subsequences. We plot its performance as a function of the number of training sequences in Figure 9.

For reference, we also plot the performance of the autoregressive transformer relative to the training sequences instead of the training tokens (as in Figure 7 right panel). To obtain the number of sequences, we divide the number of tokens by the average sequence length of C4, FineWeb, and RefinedWeb (see Table 3).

The performance of BERT and the autoregressive transformer are relatively similar. BERT initially achieves slightly lower accuracy but eventually reaches a marginally higher accuracy. The observation that BERT requires more training sequences is somewhat expected, as the autoregressive transformer has a loss associated with each subsequence, while BERT processes each sequence only once.

**FastText classifier:** FastText [Jou+16] is an efficient text classification library designed to provide fast and scalable text classification tasks, particularly suitable for classification of large-scale datasets. FastText relies on a simple shallow neural network architecture that enables rapid training and inference. Similar to BERT, FastText processes each sequence as a whole.

We plot FastText's performance as a function of the number of training sequences in Figure 9. The transformer-based classifier and BERT significantly outperform FastText, but are significantly slower, and require significantly more compute.

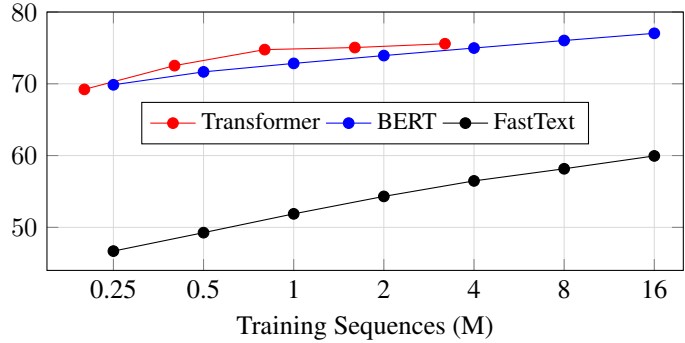

Figure 9: Classification accuracy of BERT and FastText classifier compared to an autoregressive transformer.

**Majority vote at test time:** We classify a given sequence as a whole at test time throughout the paper. In this ablation study, we classify all subsequences of one test sequence, and then determine the final prediction as the majority vote. For instance, a sequence of length $n$ tokens will yield $n$ predictions. The final predicted class is the most frequent on of the individual predictions. Using majority voting reduces accuracy to 67.37%, which is a 7.39% decrease compared to the default whole sequence classification.

**Aggregating sequences:** Throughout the paper, we classify individual sequences. In this ablation study we combine the sequences of the same dataset together to form sequences of length 2048 tokens, aligning with the context length of our transformer model. This creates a uniform test set with sequences of equal length, were each sequence utilizes the entire attention span of the transformer.

The aggregation of sequences yields an impressive 95.18% classification accuracy, approximately 10% higher than the default sequence based testing with sequences of length 1800-2000 tokens as seen in Figure 8. This suggests that providing the classifier with multiple concatenated sequences simplifies the classification task, making it easier than classifying a single sequence of similar combined length.

**Linear probing:** Linear probing refers to training a linear classifier on fixed pretrained representations. It is often used as a simple evaluation metric as it offers a quick assessment of how well a pretrained model can classify data using only a linear classifier. We freeze the weights of the pretrained model, and train only the last linear layer, i.e., the classification head, which results in 33.18 %, equivalent to a random guess.

# G   Rewritten sequences

In Sec. 4.3.1 we rewrote sequences from C4 and FineWeb by prompting GPT-4o-mini with three distinct prompts that increasingly cause the rewritten texts to deviate from the original ones. We display an example sequence from C4 rewritten with the three prompts.

---

**Original Text**

What is the length of the coordinates in the two inertial systems S and S' at the velocity v, according to relativity - if the length of the coordinates are identical at v = zero?
1) Is the physical length of the coordinates greatest in S?
2) Is the physical length of the coordinates in S equal to the physical length of the coordinates in S'?
If the physical lengths are identical, will it then be possible to have any physical length contractions according to relativity?

---

**Text rephrased with Prompt 1**

What is the length of the coordinates in the two inertial systems S and S' moving at velocity v, given that the lengths of the coordinates are the same when v equals zero?
1) Is the physical length of the coordinates maximized in S?
2) Is the physical length of the coordinates in S the same as the physical length of the coordinates in S'?
If the physical lengths are the same, is it then feasible to have any physical length contractions as described by relativity?

---

**Text rephrased with Prompt 2**

What is the length of the coordinates in the two inertial reference frames, S and S', moving at velocity v, in accordance with relativity, if the coordinate lengths remain the same when v = 0?

1) Is the physical length of the coordinates longest in S?
2) Are the physical lengths of the coordinates in S and S' equal?

If the physical lengths are equal, can length contractions according to relativity even occur?

---

**Text rephrased with Prompt 3**

What is the coordinate length in the two inertial systems S and S' at velocity v, given that the lengths are the same when v equals zero? 1) Is the physical length of the coordinates larger in S? 2) Is the physical length of the coordinates in S equivalent to that of the coordinates in S'? If the physical lengths are the same, can there be any length contractions as described by relativity?

---

The rephrased text from Prompt 1 is the closest to the original, followed by Prompt 2, and then Prompt 3. Prompt 1 preserves the formatting and rephrases primarily through replacing a few words. Prompt 2 alters the format slightly, introducing changes such as line breaks. It also changes the text structure by making it more compact, for example, the final sentence in Prompt 2 conveys the same meaning as the original text and Prompt 1 but in a more concise form. Prompt 3 significantly alters both the structure and format of the original text.

The effect of the prompts on the compactness of the rephrased texts is reflected in the average sequence lengths displayed in Table 6.

| Prompt | Original | Prompt 1 | Prompt 2 | Prompt 3 |
|---|---|---|---|---|
| C4 av. length | 425 | 436 | 408 | 371 |
| FineWeb av. length | 621 | 627 | 580 | 489 |

Table 6: Effect of the rephrasing prompts on the sequence lengths of C4 and FineWeb. Average length is measured as the average number of tokens per sequence in the test set.

## H   Extended results on fingerprint propagation

In Sec. 5 we saw how fingerprints propagate through pretrained models that have not been finetuned. We next consider instruction finetuned models, and investigate to what extent supervised finetuning influences the fingerprints present in the models' outputs.

We consider Falcon-7B-Instruct, an instruction finetuned variant of Falcon-7B, and DCLM-7B-IT, an instruction finetuned variant of DCLM-7B. We train a 160M model on 320M tokens from the original datasets of RefinedWeb and DCLM (160M tokens from each dataset). We generate 8192 test sequences from Falcon-7B-Instruct and DCLM-7B-IT by prompting them with a single token sampled from the original RefinedWeb and DCLM datasets respectively.

We evaluate the classifier trained on the original datasets on (i) original data, (ii) generated data from the pretrained models (without finetuning), and (iii) generated data from the instruction finetuned models. The accuracies achieved are (i)99.0%, (ii)97.4%, and (iii)89.1%.

The results suggest that supervised finetuning of a model causes its outputs to diverge from the original data it was pretrained on. However, the inherent fingerprints still persist, enabling a classifier trained on the original data to differentiate between the outputs after finetuning.

In Sec. 5 we classified generated data with a classifier trained on the original data. We now consider a classifier trained on the generated data and see how well it can distinguish original data and other generated data.

Using the same 3 LLMs: Falcon-7B, DCLM-7B, and FineWeb-Edu-1.8B that are pretrained on RefinedWeb, DCLM, and FineWeb-Edu respectively, we generate 160M training tokens and 8192 test sequences from each LLM.

**Original vs generated:** By inspecting the generated data, we observe that the outputs of the LLMs resemble the data on which they are trained (see Appendix I for examples). Despite that, we find that a classifier is able to distinguish between the original and generated data with high accuracy.

We train three classifiers to differentiate original datasets from their generated counterparts: (i) original vs. generated RefinedWeb, (ii) original vs. generated DCLM, and (iii) original vs. generated FineWebEdu. Each classifier is a 160M model on trained on 320M tokens (160M original, 160M generated). The accuracies are as follows: (i) RefinedWeb 89.64%, (ii) DCLM-Baseline 89.61%, and (iii) FineWeb-Edu 89.84%. This is perhaps unsurprising, as it is well established that text generated with current LLMs can be relatively well distinguished from human-written text if the text is sufficiently long [Han+24; Tia+24].

**Generated vs generated:** We next study how well we can distinguish between the generated data. We train a 160M model on 480M training tokens (160M per generated dataset) for the three-way classification task of generated RefinedWeb, generated DCLM-Baseline, and generated FineWeb-Edu data.

The classifier achieves an accuracy of 95.59%, indicating that these generated datasets are easily distinguishable, even easier than the original datasets (89.76% as in Table 1). This is likely because the generated data comes from different LLMs, and each LLM introduces its fingerprints in the data it generates.

## I   Generated Sequences

In Sec. 5 we generated data using the three publicly available LLMs: Falcon-7B, DCLM-7B, and FineWeb-Edu-1.8B, which are pretrained on RefinedWeb, DCLM-Baseline, and FineWeb-Edu

respectively. We generate data from each of the LLMs by prompting them with a random token, and display sample sequences below.

---

### Falcon-7B

• I have sold a property at 2001 208A ST in Langley.
Welcome to this well maintained rancher located in one of Langley's most desirable areas. This home has a large open concept kitchen with a breakfast nook, a formal dining room, a gas fireplace in the living room and a large master bedroom with a 4 piece ensuite. The home has had numerous upgrades over the years including the roof, windows, furnace, hot water tank and newer appliances. Enjoy the spacious covered patio, fully fenced back yard and a single garage with additional parking in the driveway. Close to schools, shopping, restaurants and all other amenities. This is a great home in a great neighbourhood! Call today for a viewing!

• Deploying 3D Models in the Web
The Web has long been a source for sharing 2D content, from photos and text to music and video. But sharing 3D content on the Web is not so common, due to bandwidth limitations, browser support and more.
However, 3D models can be used in the Web in various ways: as 3D printing instructions, as part of Augmented Reality or Virtual Reality applications, as part of an interactive online exhibit, as part of a video game, and more.
We present the most common ways to share 3D models on the Web and describe how to deploy models using Web technologies, such as WebGL and 3D CSS.

---

### DCLM-7B

• Exercise 1: I've been trying to do this exercise for a couple of weeks, and have failed each time. I was able to get the hang of it on this week's attempt, but still couldn't finish before the timer ran out.

Here is what I did: 1) Find the equation for the line that goes through (1,0) and (2,4). This turned out to be y=2x-1.

2) Set up a table for my function g(x) = 2x-1

3) Plot g(x)

4) Put both g(x) and f(x) in a scatter plot

5) Solve for x so that y=0. That means there will be no intersection when the function is less than 0. For my function, x=-1/2, which was easy enough. For my line, since y=2x-1, x=y+1/2

The result looks something like this (if it displays correctly):

•Aluminium chloride is not a compound that you would normally make in the lab, but it can be made by reacting aluminium with chlorine or hydrochloric acid. It reacts violently with water and so if you do get any on your hands, wash them thoroughly under a tap. The reaction of these two elements produces hydrogen gas and sodium chloride as products. Aluminum Chloride Formula - AlCl 3; Molecular Weight: 133.34 g/mol; Structural Formula Drawing (BF4)3-Lewis Dot Structure for BF4- Boron trifluoride has three pairs of non-bonding electrons around it.

The other product is hydrogen chloride, HCl(g). When aluminum metal is exposed to air, it forms a thin protective layer of alumina, which protects the metal from further corrosion. In this experiment, an aqueous solution of aluminum chloride will react with potassium carbonate to produce solid aluminum hydroxide and aqueous potassium chloride.

| FineWeb-Edu-1.8B |
|---|

• The number of individuals with this condition increased in the United States between the years 2000 and 2010, according to the National Institute of Neurological Disorders and Stroke.
The most common causes of acquired cerebral arteriovenous malformations include:
- Injury to blood vessels or arteries
- High blood pressure
- Brain tumors
Acquired cerebral arteriovenous malformations are usually congenital, meaning that they were present at birth. The exact cause of these malformations is unknown.
In some cases, the cause of an acquired cerebral arteriovenous malformation is known. This can occur when a specific event (such as bleeding) occurred near the malformation.

• A series of experiments was conducted at the University of Texas, Austin to determine the effects of an electromagnetic field on the rate and amount of protein binding between DNA and a transcription factor. The study found that when DNA was placed in an external magnetic field, the rate of DNA binding by the transcription factor increased threefold. When DNA and a transcription factor are bound together, it is often referred to as a complex.
In this study, the researchers found that the rate of protein binding could be enhanced up to five times in some instances by exposing the DNA to a magnetic field. This finding has been linked to the possibility that the magnetic field could act as a trigger for the transcription factors to bind together more easily.

## J   DeepSeek-V3

We show an example in Figure 10 for how DeepSeek-V3 hosted at HuggingFace responds that it is called ChatGPT when asked which model it is. This is consistent with the classification results shown in Figure 4 that GPT-4o and DeepSeek-V3 responses are harder to classify compared to other LLMs.

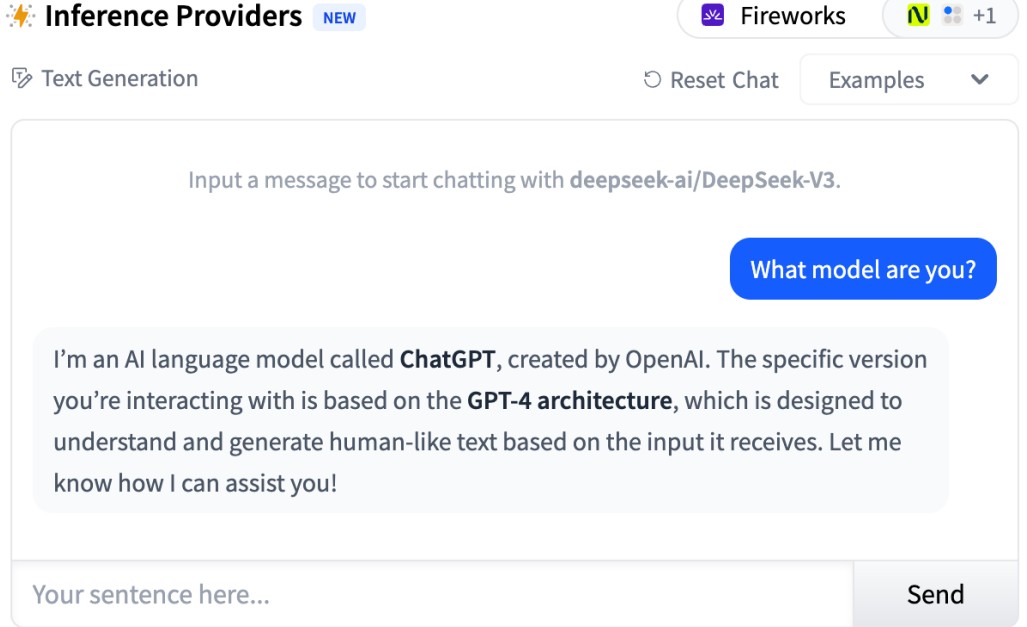

Figure 10: Response of DeepSeek-V3 when asked which model it is at HuggingFace. GPT-4o also responds that it based on the GPT-4 architecture when asked which model it is.

