# OpenReview forum: "Measuring Fingerprints of Web-filtered Text Datasets and Fingerprint Propagation Through Training"
_NeurIPS.cc/2025/Datasets_and_Benchmarks_Track — NeurIPS 2025 Datasets and Benchmarks Track spotlight_

### Official Review · Reviewer_jG3r · 2025-06-23

**Rating:** 6
**Confidence:** 4

**Summary:**

This work describes an analysis of popular CommonCrawl-based "curated datasets" (C4, Dolma, etc.) with a focus on trying to identify and understand the "fingerprints" produced by the curation choices associated with each individual dataset's curation process. A number of experiments are provided to both substantiate the existence of such fingerprints and the discuss the likely practical impacts for model trainers and guide "mitigations" that might improve generalization in the future.

**Dataset Code Accessibility:**

Yes

**Dataset Code Comments:**

The authors works exclusively with already available datasets. The core experiments use OpenLM (MIT License). The authors attach their experimental code. I inspected their repo for organization and existence of code mapping to experiments, but I did not run or directly replicate any experiments.

**Ethical Comments:**

No concerns; see above comments about "it might be nice to fit in some discussion on implications for ntellectual property / data consent  if possible".

**Ethical Considerations:**

No, there are no or only very minor ethics concerns

**Final Justification:**

Rebuttal addresses minor concerns.

**Limitations Weaknesses:**

Some limitations and opportunities for improvement:
- The paper highlights similar findings in the context of computer vision. Discussing the overlap in these results in more depth might be useful (but is reasonable to keep as is, given space constraints).
- Some design decisions could use slightly more justification (though appendix helps), e.g. choice of classifier (line 106), choice to use 7 datasets (reasonable, but any particular threshold?), choices around filtering (line 115)
- Very minor: I think this work could benefit from a short discussion on the implications for ongoing intellectual property / data consent debates, though the choice to focus in the current draft on fitting in as many experimental details in the main body seems reasonable (there's no obvious "cut" in my mind).
- Similarly, it might be also helpful to discuss at length the implications for model developers who might want to add or avoid distillation because of its interactions with fingerprinting, and generally discuss practical cases where developers might want to run experiments like this

**Strengths Contributions:**

At a high-level, this work combines a very strong conceptual contribution with a very exhaustive set of experiments (these experiments both substantiate the core claim, most importantly, and also provide detail that will be relevant to some readers, such as those who might want to perform model curation or those who just want to use CommonCrawl variants). Overall, I think this is impactful work that very much aligns with the D&B call.

To outline other specific strengths that appeared through the submission:
- Communication in the paper is strong throughout. Abstract is extremely clear and in general I think readers will appreciate the organization of the paper
- While the "headline result" here is interesting on its own, the paper does a good job of highlighting potentially non-obvious implications
- In general, the methods are cleanly described and likely will be easily replicable for many parties (see some comments below about adding a bit more justification for some minor choices)
- The main "risk" with the experiments here is that there may be a factor that was not accounted for, but overall the paper suggests a very thorough effort to mitigate this / obtain robustness (Section 4.2). I expect readers will find this convincing.
- Useful and detail Appendix.

---

> ### Author Rebuttal · Authors · 2025-07-30
>
> Many thanks for the positive feedback and assessment.
> &nbsp;
> Regarding Limitations Weaknesses:
>
> - Regarding the connection to similar findings in computer vision: While we discuss these findings in the introduction (lines 18–31) and related work (lines 75–83), we agree that a more in-depth discussion would be valuable. As the reviewer rightly noted, space constraints may limit how much can be included in the main text, so we will include a more detailed discussion in the appendix of the final paper.
>
> - We thank the reviewer for the comment regarding slightly more justification of certain design decisions (i.e., classifier, datasets). We currently provide additional justification for these choices in Appendix C and G. We fully agree that including some of this justification in the main text would improve clarity, and if accepted, we will do so in the final paper, where one additional page is allowed.
>
> - Implications for ongoing intellectual property / data consent debates: Thanks for suggesting to discuss this point. Our experiments demonstrate that fingerprints exist in text datasets, and persist through training and finetuning. Therefore, when a model is trained via distillation from outputs generated by a proprietary LLM (e.g., through the OpenAI API), it can generate data that closely mimics the proprietary LLM, even if only some of the finetuning data is generated that way. In such a situation, the resulting model may be considered to violate consent or intellectual property under usage terms. For example, OpenAI’s API terms prohibit using the API  to develop AI models that compete with OpenAI’s products. There have been concerns, for instance, that models like DeepSeek-V3 may have been partly trained or finetuned on data from OpenAI APIs (see Figure 10 in Appendix K). Our classification experiments provide empirical evidence that supports this possibility, moving the discussion beyond speculation.
>
> - Implications for model developers:  When distilling from a teacher model, our work provides a practical verification tool; by applying dataset classification experiments like ours, i.e., comparing outputs from the distilled model and the teacher LLM under matched prompts, developers can inspect how closely their trained model mimics the original. If a classifier can distinguish between them only slightly better than chance, it suggests the model may be too similar and at risk of violating data consent or intellectual property, and would require additional data diversification or permissions. In contrast, high separability suggests meaningful differentiation and offers evidence that the model is sufficiently distinct, thus mitigating consent and proprietary concerns.
> We will add these discussions to our final paper and thank the reviewer for suggesting them.
> &nbsp;
> We hope we were able to address the reviewer’s concerns, and we are happy to clarify further.

---

> > ### Comment · Reviewer_jG3r · 2025-08-05
> > **Helpful rebuttal**
> >
> > Thanks to the authors for this rebuttal. My original review had only minor questions and comments, and these are addressed (ultimately, many of them stem from space issues and not confusions). I raise my score to 6.

---

> > > ### Author Response · Authors · 2025-08-05
> > >
> > > Thank you for the valuable feedback!

---

### Official Review · Reviewer_Z8eG · 2025-07-01

**Rating:** 6
**Confidence:** 5

**Summary:**

The authors take inspiration from previous work in computer vision and investigate the fingerprints of popular pre-training datasets derived from Common Crawl, showing that automated classifiers can distinguish them, as well as models trained on them.

Specific distinguishing features of each dataset are analyzed, and the classifier approach is additionally applied to popular LLMs, offering interesting insights into their training data.

**Additional Feedback:**

Some additional aspects of the paper that could be clarified, along with general questions:
- In Section 4, where datasets are classified into 3 different categories, the chosen criteria for categorization (in particular wikipedia perplexity vs classifiers for Category 2 vs Category 3) is not sufficiently justified. In particular, RedPajama-V2 is only lightly filtered by default (filters are used but mostly to create annotations), significantly less than datasets in Category 1; and neither DCLM-Baseline nor FineWeb-Edu are filtered with Wikipedia perplexity scores (unlike what is stated on line 138).
- Lines 146-149: it is claimed that high accuracy scores in comparisons between DCLM-Baseline+FineWeb-Edu and other datasets are not surprising given that these sequences are distinct (in what seems to allude to the filtering process of these datasets). Further ahead, on lines 188-192 and in the formatting experiments it is then shown that one of DCLM’s main defining characteristics is the use of \n\n. Given this fact, the previous statement seems somewhat inconsistent (judging by Appendix D’s results those comparisons were likely driven overwhelmingly by DCLM, and not due to the criteria used to group DCLM with FineWeb-Edu, i.e., the use of quality classifiers).
- Lines 263: Why would a focus on educational texts be consistent with a smaller token budget? SmolLM2, another model trained on FW-Edu was trained for 11T tokens, for instance.
- Lines 339-340: typically the model that is “distilled” (from) is the teacher model, GPT-4o in this case, and not Qwen
- Appendix B.1: what dataset(s) is the 160M classifier pre-trained on? Is the same base model used for all experiments or is pre-training performed on different datasets for each experiment/comparison?
- Appendix D: Do the authors have any intuition for the low accuracy between FW-Edu and RP2, which is even lower than between FW vs FW-Edu (where FW-Edu is a strict subset of FW)?
- Figure 7: the axis scale makes the small difference (line 639) seem significant
- General comment about Figure 1: the plot seems to suggest that Dolma CC, RP2, and DCLM have very similar topic distributions (in contrast to FW-Edu, which was grouped with DCLM)

Overall the paper is technically solid, with interesting and relevant findings. I will raise my current score (to 6, strong accept) if the two main limitations pointed out above are addressed.

**Dataset Code Accessibility:**

Yes

**Ethical Considerations:**

No, there are no or only very minor ethics concerns

**Final Justification:**

The main findings of the work are novel and significant: it is shown that automated classifiers can successfully distinguish between documents from different Common Crawl based datasets (Section 4) and certain features affecting the accuracy of this classification are studied in Section 4.2. The classifier setup is carefully ablated, and the different experiments to test the impact of different features are sound. The main idea of the work is further explored in Sections 5 and 6 with interesting and relevant results.

The paper is well-written, well-organized, and technically solid. My concerns were appropriately addressed by the authors.

**Limitations Weaknesses:**

This reviewer finds two main weaknesses in this work.

In Section 4.2 distinguishing features of each dataset are analyzed. Specific experiments study the impact of formatting and word distributions of different datasets, and in Figure 1 the thematic distribution of each dataset is displayed. There seems to be a clear distinction between sample level characteristics (such as formatting, where for the same webpage one website could have more \n than another) and distributional/sampling aspects, where due to the content distribution of each dataset a given sample is more likely to be classified as one or another dataset. However, this distinction is never clearly stated, and particularly when classifying between FineWeb vs FineWeb-Edu further exploration of the distributional aspect would strengthen the paper, as FineWeb-Edu is strictly a subset of FineWeb (hence, without any formatting or other sample level differences but uniquely with distributional ones). In other words, every single document in FineWeb-Edu is also part of FineWeb, which raises questions that could be addressed with additional exploration of the distributional aspect of the distinguishing features.

The other main weakness is structural. While the authors state their main findings and their implications in the Introduction, the paper lacks a Conclusion section, ending abruptly after finetuning experiments. It would strengthen the paper to explicitly restate its main findings and contributions at the end, and particularly their significance and potential impact.

**Strengths Contributions:**

The main findings of the work are novel and significant: it is shown that automated classifiers can successfully distinguish between documents from different Common Crawl based datasets (Section 4) and certain features affecting the accuracy of this classification are studied in Section 4.2. The classifier setup is carefully ablated, and the different experiments to test the impact of different features are sound. The main idea of the work is further explored in Sections 5 and 6 with interesting and relevant results.

The paper is well-written and well-organized for the most part.

---

> ### Author Rebuttal · Authors · 2025-07-30
>
> We appreciate the reviewer’s willingness to raise the "current score (to 6, strong accept)", and thank the reviewer for acknowledging that the "main findings of the work are novel and significant", "the classifier setup is carefully ablated", "the paper is well-written and well-organized", and "the paper is technically solid, with interesting and relevant findings".
> &nbsp;
> Regarding Limitations Weaknesses:
>
> - Weakness 1: Thanks for highlighting the difference of sample-level differences (such as newlines \n) and distributional features (such as topic prevalence). We now made this difference more explicit in the introductory paragraph of Section 4.2. Indeed, since FineWeb-Edu is a subset of FineWeb, the distinguishability relies on distributional difference. Following the reviewer’s suggestion to further explore the distributional differences between FineWeb and FineWeb-Edu, we conducted the following three new experiments:
>
>     - Vocabulary divergence: We computed log-odds ratios for word frequencies across the two datasets. FineWeb-Edu’s most distinct words are academic and scientific terms such as doi, microbial, volcanoes, paradigm, and larvae, whereas FineWeb's most distinct words are related to entertainment and commerce, including gameplay, casino, nfl, itunes, and rental. This confirms that a distinction arises from thematic content. A long list of distinctive words for each dataset is added to the appendix.
>
>     - Semantic similarity: We measured the cosine similarity between documents using Sentence-BERT embeddings. The within-dataset similarities for FineWeb (0.054) and FineWeb-Edu (0.044) are higher than the cross-dataset similarity between FineWeb and FineWeb-Edu (0.029), confirming their distributional divergence.
>
>     - Stylistic analysis: We compared part-of-speech (POS) tag distributions to capture stylistic tendencies. FineWeb-Edu documents contain a higher proportion of nouns (19.67% vs. 18.54%) and adjectives (7.20% vs. 6.41%), characteristic of formal and descriptive writing. For instance the adjective "significant" appeared very often in FineWeb-Edu, a characteristic of academic texts. FineWeb has more pronouns (5.48% vs. 4.08%), consistent with more conversational and informal web content.
>
> - Weakness 2: Following the reviewer’s request, we have added the following conclusion section with the main findings, contribution, and significance:
>
>     "In this work, we demonstrated that popular web-filtered pretraining datasets possess unique and measurable fingerprints, despite their similar origins and curation methods. Through extensive classification experiments, we showed that neural networks can identify a sequence's source dataset with surprisingly high accuracy, a task at which humans perform poorly. We identified that these fingerprints stem from subtle distinctions in formatting, vocabulary, and content distributions. Moreover, we found that these fingerprints propagate through pretraining and finetuning. The fingerprints and their propagation can have negative consequences, for instance implications on generalization, which we show can be improved by training on a mixture of datasets. Nevertheless, they can be leveraged to estimate undisclosed pretraining data mixture proportions, and to offer insights into potential finetuning data sources."
> &nbsp;
> Regarding Additional Feedback:
>
> - Categorization criteria: Category 1 includes standard preprocessing steps such as heuristic filtering and deduplication; these steps are common across all categories. Category 2 additionally includes relatively light filtering (i.e., relatively few sequences are filtered out) based on Wikipedia perplexity scores. Category 3, in addition to Category 1 steps, incorporates relatively strong machine learning-based text filtering. We have corrected the typo in line 138 (that incorrectly stated that Category 3 includes Category 2 steps), thanks for pointing it out.
>
> - Lines 146-149: We agree with the reviewer that those specific results are mainly driven by DCLM, and have therefore modified the statement in lines 146-149 to reflect that.
>
> - Line 263: We also agree with the reviewer that LLMs trained on educational texts can have a large token budget, and have updated the statement in line 263.
>
> - Appendix B.1: The classifier is pretrained on C4 data, which is disjoint from the C4 data used for classification in all other experiments. The same pretrained classifier is used for all experiments. In the appendix in lines 658-663, we provide an ablation study for training a randomly initialized classifier (not pretrained), which only slightly reduces accuracy. Thus pretraining the classifier is not essential for the results.
>
> - Appendix D: One potential reason for the low accuracy between FW-Edu and RP2 relative to other datasets, is that RP2 was filtered using Wikipedia based perplexity scores with certain thresholds and constraints, that might bias the dataset towards the style of well-structured, formal, and informative content, which overlaps with the style of educational content targeted by FW-Edu’s machine learning-based filtering.
>
> &nbsp;
>
> We appreciate the reviewer’s valuable suggestions and additional feedback, which helped improve the paper significantly. We have fixed the typos, and added the extra experiments and the conclusion to the paper’s final version.
>
> &nbsp;
>
> We hope we were able to address the reviewer’s concerns, and we are happy to clarify further.

---

> > ### Comment · Reviewer_Z8eG · 2025-08-04
> >
> > I appreciate your detailed clarification, and will update my score to 6.

---

> > > ### Author Response · Authors · 2025-08-04
> > >
> > > Thank you for the feedback and the appreciation!

---

### Official Review · Reviewer_CdaZ · 2025-07-03

**Rating:** 6
**Confidence:** 3

**Summary:**

This work presents experiments designed to assess the capabilities of models in determining the original dataset when given rewritten text. In other words, it tests models’ ability to attribute input data. The ability of models to classify data means that it's possible to identify when and where tampering occurs during pre-processing. Models perform better than humans in tracing back datasets to outputs in these mixtures.

**Dataset Code Accessibility:**

Yes

**Ethical Considerations:**

No, there are no or only very minor ethics concerns

**Final Justification:**

All issues were resolved and extensions of the analyses were conducted, so the score is being increased by 1 (to a 6).

**Limitations Weaknesses:**

This work could have been conducted on more models, such as comparing open-source models to the 4o-mini model they use for their experiments. As a baseline, it would have been valuable to include a novel dataset or a dataset not currently on the web as a baseline, and to consider the other contributors to classification accuracy.

**Strengths Contributions:**

The study is expansive and covers the largest web-scraped datasets. They categorize their seven datasets by their filtering techniques (i.e., language-filtered, heuristically filtered, and deduplicatively filtered). Given different combinations of the dataset types, they study the classification accuracy of models. They compare this to human capabilities, which is a strength of this paper, providing a baseline of performance on the tasks outlined.

---

> ### Author Rebuttal · Authors · 2025-07-30
>
> Many thanks for the positive assessment and for noting that "the study is expansive and covers the largest web-scraped datasets".
> &nbsp;
> Regarding Limitations Weaknesses:
>
>
> - Following the reviewer’s suggestion to "include a novel dataset or a dataset not currently on the web", we created a synthetic dataset by prompting several LLMs (Qwen, Llama, DeepSeek, ChatGPT, Claude, and Gemini) with prompts from OpenHermes-2.5 and Alpaca. We collected the responses, shuffled them and refer to this new dataset as "Synthetic". We then performed pair-wise classification experiments (with the 160M classifier) between the newly generated dataset, and the seven main datasets considered in the paper. The results are reported in the table below. The synthetic dataset is highly distinguishable from the web-filtered datasets.
>
>     |       | C4 | FineWeb | RefinedWeb | DolmaCC | RedPajama | DCLM | FineWeb-Edu |
>     |:----:|:----:|:----:|:----:|:----:|:----:|:----:|:----:|
>     | Synthetic | 95.3% | 96.1% | 95.8% | 96.1% | 96.0% | 95.4% | 96.5% |
>
> - We appreciate the reviewer’s request to compare more open-source models to the 4o-mini model we used for our experiments. In the paper, we used the 4o-mini model for the rewriting experiments in Section 4.2.1, and the 4o model for the classification experiments in Section 6.
>     - Regarding the classification experiments, we compared three open-source LLMs (Llama, Qwen, DeepSeek) to 4o in Section 6. Following the reviewer’s request, we also compared these to 4o-mini by performing pair-wise classification experiments with the 160M classifier. The accuracies are:
>
>           4o-mini vs Qwen-2.5-72B: 73.3%
>           4o-mini vs Llama-3.3-70B: 81.3%
>           4o-mini vs DeepSeek-V3: 75.6%
>
>         As seen from the results, 4o-mini is generally distinguishable from these open-source models.
>
>
>     - Regarding the rewriting experiments, we followed the reviewer’s request, and repeated the rewriting experiment in Section 4.2.1 with the open-source model Qwen2.5-14B instead of 4o-mini. We achieved the following accuracies:
>
>           Prompt 1: 82.0%
>           Prompt 2: 77.8%
>           Prompt 3: 69.8%
>
>         The accuracies are similar to those obtained with 4o-mini (lines 209,210), highlighting that the fingerprints persist in rewritten text regardless of the model used for rewriting.
> &nbsp;
> We hope we were able to address the reviewer’s concerns, and we are happy to clarify further.

---

### Official Review · Reviewer_MN21 · 2025-07-05

**Rating:** 5
**Confidence:** 2

**Summary:**

This paper investigates the "fingerprints" in large-scale text datasets used for LLMs. The author first demonstrates that the datasets possess distinguishable fingerprints and can be detected in the outputs of LLMs via dataset classification. Moreover, the authors also explore the impact of fingerprints on model generalization and their potential use in estimating the composition of pretraining data mixtures.

**Additional Feedback:**

See weakness.

**Dataset Code Accessibility:**

Yes

**Dataset Code Comments:**

The author provides sufficient evidence to support reproducibility.

**Ethical Considerations:**

No, there are no or only very minor ethics concerns

**Final Justification:**

I would like to keep my positive score since most of my concerns have been resolved.

**Limitations Weaknesses:**

1. The authors randomly sample 100,000 sequences from each dataset to analyze sequence length statistics. Does it will induce potential biases to the results? In other words, the author should conduct a sensitivity analysis to assess how different sample sizes or sampling methods affect the results.

2.It is unknown whether the fingerprint propagation effect observed in models like Falcon-7B can be applied to more advanced, state-of-the-art LLMs such as Qwen3 or Gemma3.

3. The paper demonstrates that sequences are distinguishable across datasets with high classifier accuracy. Could such distinguishability be leveraged to detect specific pretraining data used in LLMs [1 ]?

[1] Shi W, Ajith A, Xia M, et al. Detecting pretraining data from large language models. ICLR, 2024.

**Strengths Contributions:**

1. The research focuses on an important issue in LLMs: the presence and impact of dataset-specific fingerprints.

2 . The experiments are comprehensive. The authors conduct experiments on a variety of popular, open-source datasets derived from CommonCrawl and include popular LLMs like ChatGPT-4o, Qwen-2.5-72B and DeepSeek-V3.

3. The results are promising. The authors claim that the sequences from popular pretraining text datasets can be well classified, which is well-supported by the experiments shown in Section 4.

---

> ### Author Rebuttal · Authors · 2025-07-30
>
> Many thanks for mentioning that "the research focuses on an important issue in LLMs", "the experiments are comprehensive", and "the results are promising".
> &nbsp;
> Regarding Limitations Weaknesses:
>
> 1. Following the reviewer’s suggestion to conduct a sensitivity analysis with different sample sizes, we analyzed the variability of mean sequence lengths within a 100k sequence sample. Specifically, we shuffled the sample and divided it into 10 disjoint bins of 10k sequences each, and computed the mean sequence length for each bin. The table below reports the average ± standard deviation of these bin-level means:
> | C4 | FineWeb | RefinedWeb | DolmaCC | RedPajama | DCLM | FineWeb-Edu |
> |:----:|:----:|:----:|:----:|:----:|:----:|:----:|
> | 475±8 | 705±14 | 621±10 | 838±17 | 1107±20 | 1220±24 | 1071±19 |
>
>     The standard deviation is under 2% of the corresponding mean value in most cases, indicating low variability and stable estimates.
>
>
>
> 2. Regarding whether the fingerprint propagation effect observed in models like Falcon-7B can be applied to more advanced, state-of-the-art LLMs such as Qwen3 or Gemma3:
>     - Unfortunately, the pre-training data of Qwen3 and Gemma3 is not public, and therefore we can’t conduct the same experiment on them as we did in Section 5. Our fingerprint propagation experiment focuses on publicly available LLMs pretrained on a known  dataset, such that we can classify the generated data with a classifier trained on the original data. Falcon-7B, for instance, is pretrained only on RefinedWeb (see Section 5 lines 253-263).
>      - However, in Section 6 we extensively study the distinguishability of state-of-the-art LLMs including Qwen, Llama, DeepSeek, ChatGPT, Claude, and Gemini, and with the experiment explained in Section 6, they are distinguishable.
>
>
> 3. Regarding detection of the pretraining data used in LLMs, existing works such as the one referenced by the reviewer [1] rely on the probability assigned by the LLM to a given sequence to determine whether it was likely seen during pretraining. In contrast, our classification experiments operate at the dataset level rather than focusing on individual sequences, aiming to infer whether a dataset contributed to pretraining and to also estimate its relative proportion within the overall pretraining mixture. For instance, see Figure 2c in Section 5.1, where the proportions were closely estimated, and no sequences were classified as Arxiv or StackExchange (SE) confirming the LLM was not pretrained on them.
>
>     [1] Shi W, Ajith A, Xia M, et al. Detecting pretraining data from large language models. ICLR, 2024.
> &nbsp;
> We hope we were able to address the reviewer’s concerns, and we are happy to clarify further.

---

> ### Comment · Reviewer_MN21 · 2025-08-05
>
> Thank you for your response. Most of my concerns have been resolved, and I raise my score to 5.

---

### Decision · Program_Chairs · 2025-09-18

**Decision:**

Accept (spotlight)

**Comment:**

I thank the reviewers for their thoughtful and detailed reviews and for engaging in discussion with the authors.

The paper was recognized for the breadth of experiments, the care in their design, and the findings it reports.

The reviewers’ concerns included the need for more sensitivity analyses of sampling methods and sequence lengths (MN21), the limited exploration of newer state-of-the-art models (MN21), and the distinction between sample-level versus distributional dataset differences (Z8eG), as well as other suggested experiments (CdaZ). The absence of a conclusions section and deeper discussions were also noted (Z8eG, jG3r). The additions offered by the authors in their reply seemed to resolve most of the reviewers' concerns.

The final recommendations were strong accept (CdaZ, Z8eG, jG3r) and accept (MN21).

There were no significant ethical issues noted. The study relied only on publicly available datasets and responsibly released code.

Having read the paper, the reviews, and the discussion, I recommend accepting the paper. It offers rigorous evidence of dataset fingerprints and their propagation, and provides useful insights for the community at a time when transparency in LLM training data is increasingly important.

===== FINAL UPDATE FROM DB Track PCs ====

The final decision for this paper has been taken by the program chairs after consultation with the SACs. All Senior Area Chairs have ranked papers according to the feedback from the AC during the review process. We decided to leave the original meta-review to reflect the opinion of the AC in light of the initial discussions with reviewers and SAC.